# Unique adipose tissue invariant natural killer T cell subpopulations control adipocyte turnover in mice

Sang Mun Han[1,7], Eun Seo Park [2,7], Jeu Park[1,7], Hahn Nahmgoong[1], Yoon Ha Choi[3], Jiyoung Oh[4], Kyung Min Yim[1], Won Taek Lee [1], Yun Kyung Lee [5], Yong Geun Jeon [1], Kyung Cheul Shin[1], Jin Young Huh [6], Sung Hee Choi[5], Jiyoung Park [4], Jong Kyoung Kim [3,8] ✉ & Jae Bum Kim [1,8] ✉

Adipose tissue invariant natural killer T (iNKT) cells are a crucial cell type for adipose tissue homeostasis in obese animals. However, heterogeneity of adipose iNKT cells and their function in adipocyte turnover are not thoroughly understood. Here, we investigate transcriptional heterogeneity in adipose iNKT cells and their hierarchy using single-cell RNA sequencing in lean and obese mice. We report that distinct subpopulations of adipose iNKT cells modulate adipose tissue homeostasis through adipocyte death and birth. We identify KLRG1[+] iNKT cells as a unique iNKT cell subpopulation in adipose tissue. Adoptive transfer experiments showed that KLRG1[+] iNKT cells are selectively generated within adipose tissue microenvironment and differentiate into a CX3CR1[+] cytotoxic subpopulation in obese mice. In addition, CX3CR1[+] iNKT cells specifically kill enlarged and inflamed adipocytes and recruit macrophages through CCL5. Furthermore, adipose iNKT17 cells have the potential to secrete AREG, and AREG is involved in stimulating adipose stem cell proliferation. Collectively, our data suggest that each adipose iNKT cell subpopulation plays key roles in the control of adipocyte turnover via interaction with adipocytes, adipose stem cells, and macrophages in adipose tissue.

White adipose tissue (WAT) is a key energy storage organ that undergoes dynamic remodeling in response to fluctuations in energy states[1–3]. In obesity, unhealthy WAT remodeling occurs with expanded adiposity and immune cell infiltration, resulting in obesity-related metabolic complications including insulin resistance[4,5]. It is likely that obese adipose tissue expands in two different processes, which differ in morphology and metabolic outcomes. Hypertrophic WAT expansion often exhibits pathological phenotypes such as inflammation, hypoxia, and fibrosis, whereas hyperplastic WAT expansion relatively exhibits beneficial phenotypes with a greater ability for energy storage, contributing to improved metabolic parameters[2,6,7].

[1]National Leading Researcher Initiatives Center for Adipocyte Structure and Function, Institute of Molecular Biology and Genetics, School of Biological Sciences, Seoul National University, Seoul 08826, Republic of Korea. [2]Department of New Biology, DGIST, Daegu 42988, Republic of Korea. [3]Department of Life Sciences, POSTECH, Pohang 37673, Republic of Korea. [4]Department of Biological Sciences, College of Information and Biotechnology, Ulsan National Institute of Science and Technology, Ulsan 44919, Republic of Korea. [5]Internal Medicine, Seoul National University College of Medicine & Seoul National University Bundang Hospital, Seoul 03080, Republic of Korea. [6]Department of Life Science, Sogang University, Seoul 04107, Republic of Korea. [7]These authors contributed equally: Sang Mun Han, Eun Seo Park, Jeu Park. [8]These authors jointly supervised this work: Jong Kyoung Kim, Jae Bum Kim. ✉e-mail: blkimjk@postech.ac.kr; jaebkim@snu.ac.kr

Cellular turnover to remove impaired cells and produce healthy cells is crucial for maintaining tissue homeostasis[8]. In this regard, adipocyte turnover would be important for adipose tissue homeostasis by removing large and inflamed adipocytes and generating new small adipocytes[6,9]. Given that the number of adipocytes appears to be constant in adults[10], it is likely that adipocyte birth and death are closely associated with each other[11]. Adipocyte turnover involves complex processes wherein multiple cell types should be harmoniously coordinated. Adipocyte turnover involves the following steps: induction of adipocyte death, efferocytosis of dead adipocytes by macrophages, adipose stem cell (ASC) proliferation, and adipocyte differentiation[3]. However, it is still elusive how the adipocyte turnover process is precisely regulated.

Immune cells in adipose tissue are important regulators of adipose tissue metabolism[12]. Adipose immune cells modulate not only adipose tissue inflammation by secreting various cytokines but also adipocyte fate, including adipocyte death, clearance, and adipogenesis[13–15]. Among numerous adipose immune cells, invariant natural killer T (iNKT) cells have recently gotten a lot of attention owing to their roles in adipose tissue remodeling[16–20]. iNKT cells are a type of innate-like T cells that express semi-invariant T cell receptors (TCRs) composed of invariant alpha chains and a limited repertoire of beta chains that recognize CD1d-loaded lipid antigens[21–23]. In the early stages of development, iNKT cells appear to be recruited into adipose tissue from the thymus, and then they maintain their number and unique characteristics within adipose tissue after adolescence, probably contributing to the maintenance of adipose tissue homeostasis[24–26]. Further, it has been suggested that some adipose iNKT cells acquire tissue-specific characteristics in a lipid-rich adipose tissue microenvironment and produce anti-inflammatory cytokines[26]. In addition, adipose iNKT cells are important for the regulation of adipocyte turnover. In obesity, adipose iNKT cells facilitate the death of hypertrophic and inflammatory adipocytes and promote ASC proliferation, eventually leading to adipose tissue remodeling[14,18]. However, the heterogeneity of adipose iNKT cells and their role in orchestrating adipocyte turnover have not been fully elucidated. Moreover, how protective functions of adipose iNKT cells are conferred during the developmental process and obesity is largely unknown.

In this study, by adopting single-cell RNA sequencing (scRNA-seq) analysis accompanied with TCR repertoire analysis, we investigated the heterogeneity of adipose iNKT cells under normal chow diet (NCD) and high-fat diet (HFD) conditions. We identified adipose tissue-specific iNKT cell subpopulations, by comparing adipose iNKT cells with those from other organs. We also examined whether adipose-specific iNKT cells would be generated from iNKT cell-intrinsic factors or from adipose tissue microenvironment. Using in vivo and in vitro experimental approaches including adoptive transfer, fluorescence-activated cell sorting (FACS), and coculture, we found that adipose-specific iNKT cell subpopulations have unique roles in adipocyte turnover via crosstalk with different cell types, including adipocytes, ASCs, and macrophages. Moreover, trajectory analysis and iNKT cell lineage-tracing system identified the hierarchy among adipose iNKT cell subpopulations. Together, these data suggest that adipose iNKT cell subpopulations coordinate adipocyte fate and adipose immune responses, thereby contributing to adipose tissue homeostasis.

## Results

### scRNA-seq analysis reveals distinct subpopulations of adipose iNKT cells

Although adipose iNKT cells appear to have tissue-specific characteristics compared to iNKT cells from other organs[25,26], it is largely unknown whether tissue-specific features of adipose iNKT cells would facilitate their functions in adipocyte turnover. To address this, we performed scRNA-seq of iNKT cells sorted from epididymal WAT and compared them with iNKT cells sorted from thymus (Supplementary Fig. 1a), where iNKT cell development takes place. As shown in Fig. 1a, b, adipose iNKT cells and thymic iNKT cells were separately clustered, indicating that there would be distinct molecular features with tissue-specific gene expression: *Nfil3* (also known as E4BP4), *Klrg1*, and *Nr4a1* in WAT and *Zbtb16* (also known as PLZF) and *Il7r* in thymus. These tissue-specific gene expression profiles were similar to those from a previous report comparing splenic and adipose iNKT cells[25], implying that there would be distinct transcriptome profiles in adipose iNKT cells (Supplementary Fig. 1b–g, Supplementary Data 1).

Examining heterogeneity, adipose iNKT cells were selected and reclustered (Fig. 1c). Among the six subpopulations of adipose iNKT cells (A1–A6), most were either *Tbx21*-expressing iNKT1 cells (A1, A2, and A4–A6) or *Rorc*-expressing iNKT17 cells (A3) (Fig. 1d and Supplementary Fig. 1h)[27,28]. Most adipose iNKT cell subpopulations except A4 and A6 were transcriptionally similar to previously reported subpopulations (Supplementary Fig. 1i)[26]. As three minor subpopulations (A4–A6) composed less than 2.5% of the total adipose iNKT cells (Fig. 1e), we primarily focused on three major subpopulations (A1–A3) in further analyses. KLRG1 and Ly6C were selected as surface antigens to distinguish the three subpopulations (Fig. 1f and Supplementary Fig. 2a–d). KLRG1⁻Ly6C⁺, KLRG1⁺, and KLRG1⁻Ly6C⁻ adipose iNKT cells matched with A1, A2, and A3 subpopulations, respectively. Distinguishing adipose iNKT cells using these surface antigens successfully reflected both subtype ratio and expression of key transcription factors (Fig. 1g and Supplementary Fig. 2e).

Even though thymic and adipose iNKT cells have distinct transcriptome profiles, the origin of their tissue-specific adaptations is largely unknown. To characterize shared and tissue-specific features of adipose iNKT cell subpopulations, we performed projection analysis using thymic, splenic, hepatic, and lymph node iNKT cells[29]. iNKT cells from other organs were mostly projected on A1 subpopulation, but not on A2 subpopulation or the lower portion of A3 subpopulation (Fig. 1h–l). These data imply that A2 subpopulation and a subset of A3 subpopulation might have adipose-specific characteristics, whereas A1 subpopulation shows shared gene expression profiles across organs. In addition, tissue-specific genes of adipose iNKT cells were highly expressed in A2 subpopulation (Fig. 1b, d and Supplementary Fig. 2f–h), indicating that the A2 subpopulation might determine tissue-specific characteristics of adipose iNKT cells.

To verify whether A2 would indeed be an adipose-specific subpopulation, various iNKT cell-bearing organs were tested to determine whether they express the A2 subpopulation-specific surface marker KLRG1. KLRG1⁺ iNKT cells were barely detected in other organs, whereas approximately 30% of iNKT cells from WAT were KLRG1⁺ (Fig. 1m). To test whether this pattern would be sex- or strain-specific, we examined female C57BL/6 and male BALB/c mice. As they exhibited similar patterns to male C57BL/6 mice (Supplementary Fig. 2i, j), we named the A2 subpopulation as 'adipose-specific (As)-iNKT1'. Also, there were KLRG1⁺ iNKT cells in human omental adipose tissue (Supplementary Fig. 2k), suggesting that humans might contain analogous iNKT subpopulation. On the other hand, as KLRG1⁻Ly6C⁺ iNKT cells were abundant in other organs (Fig. 1n), we named A1 subpopulation as 'adipose universal (Au)-iNKT1'. A3–A6 subpopulations were named as 'A-iNKT17', 'adipose cytotoxic (Ac)-iNKT1', 'A-Cycling iNKT1', and 'A-interferon stimulated gene (ISG) iNKT1', respectively (Fig. 1o), following their signature gene expression profiles (Fig. 1d). Together, these data suggest that adipose tissue-selective KLRG1⁺ As-iNKT1 cells might mediate distinct features of adipose iNKT cells.

### Adipose tissue microenvironment generates As-iNKT1 cells

Peripheral iNKT cells start their maturation in the thymus, migrate into peripheral tissues via circulation, and remain in peripheral organs as long-term residents[24,30,31]. Consistent with a previous report[25], we observed that the number of iNKT cells increased with age in WAT

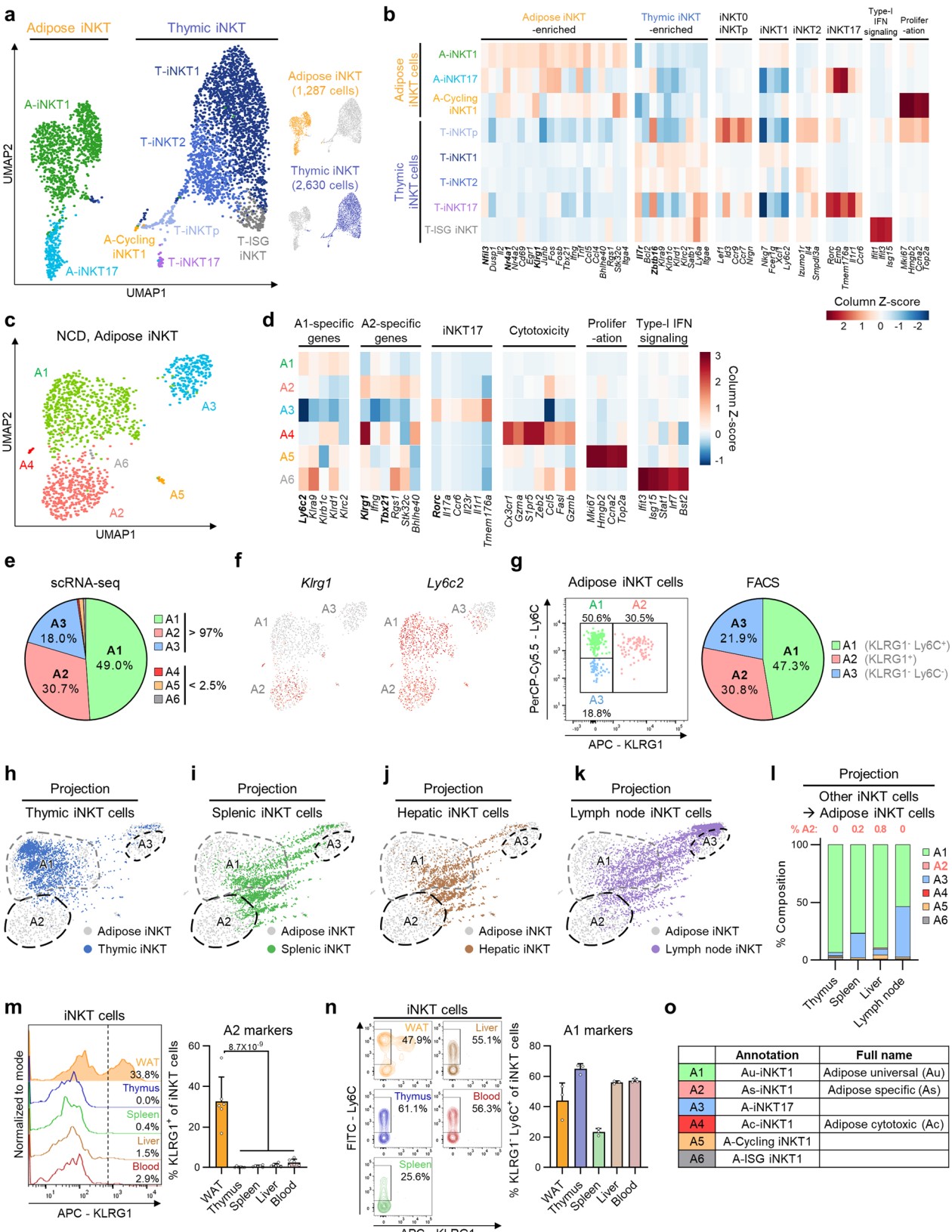

(Fig. 2a). To investigate the underlying mechanisms by which KLRG1⁺ As-iNKT1 cells would be enriched in WAT, we carefully examined when As-iNKT1 cells could emerge in adipose tissue. Only a very few As-iNKT1 cells were found in WAT until 4 weeks, whereas their proportion increased after 8 weeks (Fig. 2b). To examine the generation of As-iNKT1 cells after 4 weeks of age, we tested when

iNKT cells would infiltrate into adipose tissue during developmental process by adoptive transfer experiments using recipient mice of different ages (Fig. 2c). As shown in Fig. 2d, CD45.1⁺ iNKT cells injected at 3 weeks of age accounted for approximately 8% of the total adipose iNKT cells, and this ratio greatly decreased when they were injected at 8 weeks of age. More importantly, these data

**Fig. 1 | scRNA-seq analysis reveals distinct subpopulations of adipose iNKT cells. a** Unsupervised clustering of TCRβ^int/CD1d.PBS57 tetramer⁺ iNKT cells from 16-week-old male mice. 1,287 cells from epididymal white adipose tissue (WAT) and 2,630 cells from thymus of 16-week-old male mice on a uniform manifold approximation and projection (UMAP) plot. **b** Heatmap showing the expression levels of tissue-selective genes and iNKT cell subtype markers. **c** Unsupervised clustering of adipose iNKT cells from 16-week-old male mice on a UMAP plot. **d** Heatmap showing the expression levels of subpopulation marker genes. **e** The ratio of each adipose iNKT cell subpopulation in scRNA-seq data. **f** Gene expression levels of *Klrg1* and *Ly6c2* in adipose iNKT cells. **g** Representative FACS plot and proportion of major adipose iNKT cell subpopulations from 16-week-old male mice (*n* = 5). **h**–**k** Projection plots of iNKT cells from other organs to adipose iNKT cells.

iNKT cells from other organs are projected to average of similar adipose iNKT cells' coordinates. Adipose iNKT cells are shown in gray, and iNKT cells from other organs are shown in color. Thymic, splenic, hepatic, and lymph node iNKT cells were used. scRNA-seq data of splenic, hepatic, and lymph node iNKT cells were obtained from GSE161495. **l** Result of reference mapping of iNKT cells from other organs on adipose iNKT cells. The ratio of other iNKT cells mapped to specific adipose iNKT cell subpopulations was depicted as bar graphs and the percentage of cells mapped to A2 subpopulation was indicated on each bar graph. **m** Representative FACS plots and proportion of KLRG1⁺ iNKT cells from 10-week-old male mice (*n* = 6). **n** Representative FACS plots and proportion of KLRG1⁻Ly6C⁺ cells from 10-week-old male mice (*n* = 3). **o** Annotation table of each adipose iNKT cell subpopulation. Data are represented as mean ± SD. One-way ANOVA (**m**).

propose that the majority of adipose iNKT cells would infiltrate into adipose tissue in the early stage of life even before mice turned 3 weeks old. Thus, we speculated that the increase in As-iNKT1 cells after 4 weeks of age might be mediated by fat tissue microenvironment wherein iNKT cells reside. To test this, primary iNKT cells established from splenic iNKT cells were injected and subjected to examine the degree of KLRG1 expression according to infiltrated organs (Fig. 2e and Supplementary Fig. 3a–c). The proportion of KLRG1⁺ iNKT cells was significantly higher in WAT, indicating that adipose tissue-specific microenvironment would be prone to mediate the transition of KLRG1⁻ iNKT cells into KLRG1⁺ iNKT cells (Fig. 2f). Also, a small subset of injected Au-iNKT1 cells converted into KLRG1⁺ As-iNKT1 cells after three weeks (Supplementary Fig. 3d–f). In WAT, As-iNKT1 cells were more proliferative than other iNKT cells, which could facilitate their increase after adolescence (Fig. 2g).

Next, we identified adipose tissue-specific factor(s) involved in As-iNKT1 cell generation. To figure out adipose tissue-specific microenvironmental factor(s) upregulated after 4 weeks of age, adipocytes from 4-, 8-, and 16-week-old mice were tested for their mRNA expression profiles. Adipocytes upregulated microenvironment-associated genes, such as *Cd1d1*, *Adipoq*, and *Pnpla2*, which correspond to a lipid antigen-presenting molecule, adipokine, and lipolytic gene, respectively, as age increased (Fig. 2h). To examine whether As-iNKT1 cell generation might be mediated by chronic activation via CD1d-loaded lipid antigen(s), CD45.1⁺ iNKT cells were injected into young WT mice or CD1d KO littermates. As indicated in Fig. 2i, there were no differences observed between two genotypes in the ratio of As-iNKT1 cells among transferred iNKT cells. We then tested whether secretory factor(s) from adipocytes could affect As-iNKT1 generation. Intriguingly, adipocyte conditioned media (AD CM) upregulated several marker genes of As-iNKT1 cells, such as *Klrg1*, *Bhlhe40*, and *Rgs1* while downregulating Au-iNKT1 marker genes (Fig. 2j, k and Supplementary Fig. 3g, h). These patterns were less prominent in hepatocyte-conditioned media (Hep CM)-treated group than in AD CM-treated group (Fig. 2j, k). Thus, these data suggest that factor(s) secreted from adipocytes would mediate the generation of As-iNKT1 cells in adipose tissue microenvironment.

In adipose tissue, iNKT cells actively secrete pro-inflammatory and anti-inflammatory cytokines to modulate adipose tissue immunity[14,16,26,32]. To investigate the immunomodulatory characteristics of major iNKT cell subpopulations, including As-iNKT1, Au-iNKT1, and A-iNKT17 cells, cytokine profiles were examined. These subpopulations were distinguished by the expression pattern of KLRG1 and Ly6C after activation (Supplementary Fig. 3i). Among three major subpopulations, As-iNKT1 cells showed the highest IFNγ and TNFα production after activation (Fig. 2l–n). A-iNKT17 cells produced high levels of IL-17A and IL-4, whereas Au-iNKT1 cells showed an intermediate level of Th1- and Th2-type cytokine production (Fig. 2l–n). Thus, it seems that inflammatory tone of adipose iNKT cells might be partly determined by relative ratio of each adipose iNKT cell

subpopulation upon various metabolic stimuli, such as 1-week HFD feeding (Supplementary Fig. 3j).

To further characterize As-iNKT1 cells, we compared two major iNKT1 cell subpopulations: Au-iNKT1 and As-iNKT1 cells. KEGG pathway analysis revealed that Th1/Th2 differentiation and TCR signaling pathway were enriched in As-iNKT1 cells while NK cell-mediated cytotoxicity-related genes were enriched in Au-iNKT1 cells (Supplementary Fig. 3k). Accordingly, As-iNKT1 cells highly expressed TCR signaling components and their downstream genes, such as *Lat*, *Lcp2*, and *Nr4a1*, whereas Au-iNKT1 cells abundantly expressed activating or inhibitory NK receptors, such as *Ncr1*, *Klrk1*, *Klrc1*, and *Klrd1* (Supplementary Fig. 3l)[33,34]. Taken together, these data propose that adipose tissue microenvironment would mediate the generation of As-iNKT1 cells.

## In obesity, As-iNKT1 cells give rise to Ac-iNKT1 cells

In obesity, adipose iNKT cells play a protective role in the maintenance of adipose tissue homeostasis through clearance of detrimental adipocytes[18]. Nonetheless, it remains elusive how adipose iNKT cells would acquire or boost the roles of promoting adipocyte clearance in obesity. To investigate obesity-induced changes in adipose iNKT cells, we performed scRNA-seq paired with TCR repertoire analysis on adipose iNKT cells from NCD-, 1-week, or 8-week HFD-fed age-matched mice (Fig. 3a). Even in HFD-fed conditions, adipose iNKT cells were clustered into the same six subpopulations (A1–A6) as in NCD condition with similar marker gene expression (Fig. 3b). However, the proportions of iNKT cell subpopulations and their gene expression patterns were altered upon HFD. The proportions of As-iNKT1, Ac-iNKT1, and A-Cycling iNKT1 cells were increased after 8-week HFD feeding, whereas the proportion of Au-iNKT1 cells was relatively decreased, and those of A-iNKT17 and A-ISG iNKT1 cells were not altered (Fig. 3c and Supplementary Fig. 4a). A-Cycling iNKT1 and A-ISG iNKT1 cells were minor subpopulations accounting for less than 5% of the total adipose iNKT cells under all conditions (Fig. 3c and Supplementary Fig. 4a). Thus, we decided to exclude A-Cycling iNKT1 and A-ISG iNKT1 cells from further analyses and mainly focused on four subpopulations: Au-iNKT1 (A1), As-iNKT1 (A2), A-iNKT17 (A3), and Ac-iNKT1 cells (A4). To distinguish *Klrg1*-expressing Ac-iNKT1 cells from As-iNKT1 cells, CX3CR1 was selected as the surface antigen of Ac-iNKT1 cells (Supplementary Fig. 4b–d). FACS analysis confirmed that As-iNKT1 and Ac-iNKT1 cells were increased upon 8 weeks of HFD feeding, whereas Au-iNKT1 cells were decreased (Fig. 3d and Supplementary Fig. 4e).

To understand how obesity could modulate gene expression profiles in As-iNKT1 cells, differentially expressed genes (DEGs) between NCD and 8-week HFD-fed conditions were analyzed. Most adipose iNKT cell subpopulations showed activation-associated phenotypes such as upregulation of Nur77 (*Nr4a1*) (Fig. 3e)[35]. As-iNKT1 cells upregulated proliferation-associated genes (*Hmgb2* and *Plk3*)[36,37], their characteristic gene associated with *Ifng* expression (*Bhlhe40*)[38], and specific TCR Vβ chain gene (*Trbv13-1*,

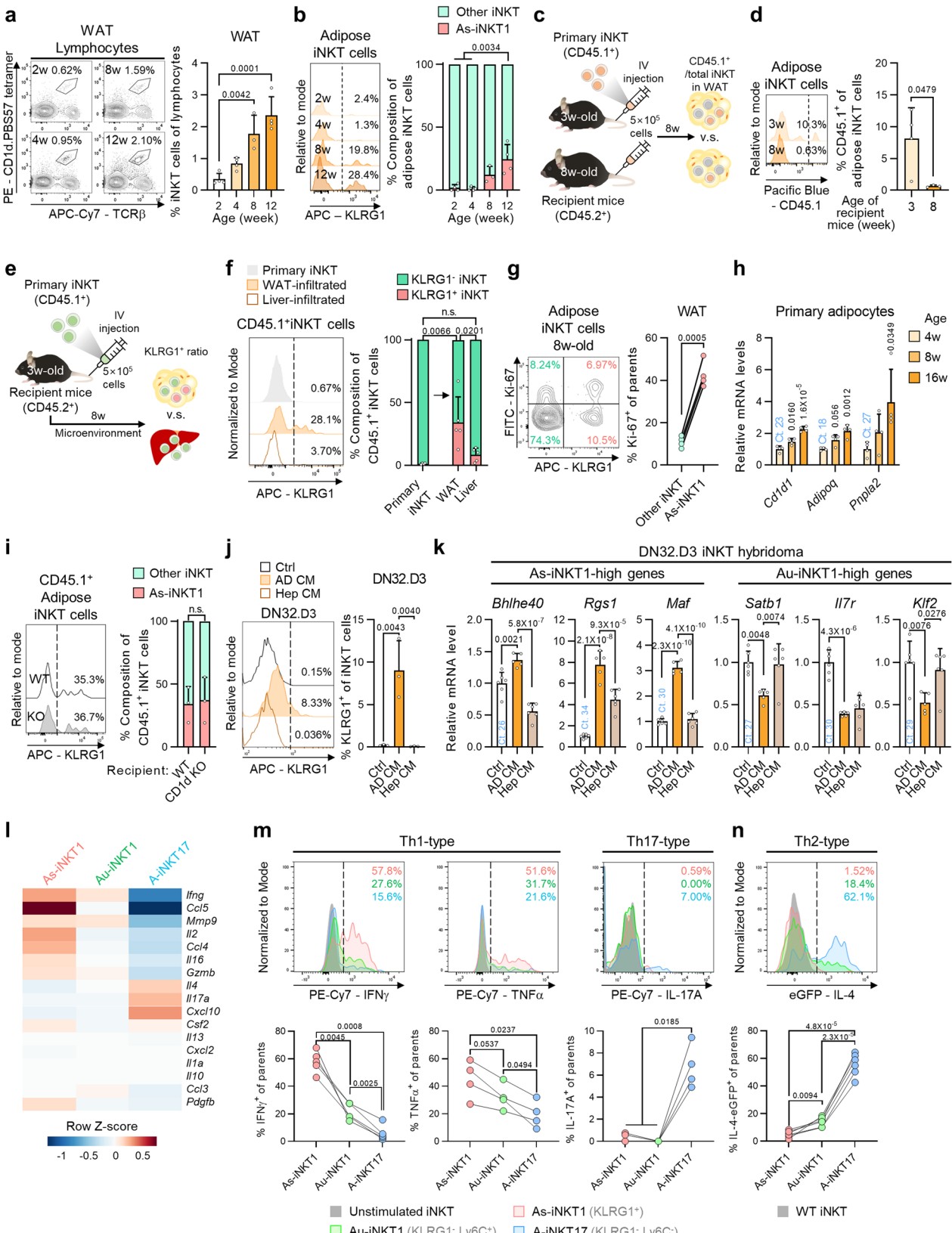

Vβ8.3) upon 8-week-HFD feeding (Fig. 3e and Supplementary Fig. 4f). On the other hand, several NK receptor genes such as *Cd160* and *Klrk1* were downregulated in As-iNKT1 cells (Fig. 3e and Supplementary Fig. 4f). These data suggest that in obesity, As-iNKT1 cells would become more proliferative and certain As-iNKT1 cells with specific Vβ might be selected. Other adipose iNKT cell

subpopulations also changed upon HFD. Au-iNKT1 cells upregulated NK receptors such as *Klrd1* and *Klrc2* while downregulating *Ifng* (Supplementary Fig. 4g). A-iNKT17 cells upregulated *Zbtb16* and *Rora*, while downregulating some GTPase of immunity-associated protein (GIMAP) family genes and MHC I molecules (Supplementary Fig. 4h).

**Fig. 2 | Adipose tissue microenvironment generates As-iNKT1 cells.**
**a**, **b** Representative FACS plots and proportion of iNKT cells (**a**) and As-iNKT1 cells (**b**) from male mice WAT (2w, 4w, 12w (n = 4), and 8w (n = 3)). **c** Experimental scheme for adoptive transfer of primary iNKT cells established from spleen.
**d** Representative FACS plot and proportion of CD45.1[+] iNKT cells among adipose iNKT cells (3w (n = 4) and 8w (n = 3)). **e** Experimental scheme for adoptive transfer of primary iNKT cells. **f** Representative FACS plot and proportion of KLRG1[+] cells among primary iNKT cells (n = 4) and infiltrated CD45.1[+] iNKT cells (n = 5).
**g** Representative FACS plot and proportion of Ki-67[+] cells from 8-week-old male mice (n = 4). **h** mRNA levels in adipocyte fraction of WAT from male mice (n = 4). p-values; versus 4-week-old. **i**, **j** Representative FACS plot and proportion of KLRG1[+] cells among CD45.1[+] iNKT cells infiltrated into WAT of WT or CD1d KO mice (n = 3) (**i**) and among DN32.D3 iNKT hybridoma cells after culture with Control media (Ctrl), primary adipocyte-conditioned media (AD CM), or primary hepatocyte-

conditioned media (Hep CM) (n = 3 biologically independent samples) (**j**). **k** mRNA levels in DN32.D3 cells after culture with Ctrl (n = 6), AD CM (n = 5), or Hep CM (n = 6). **l** Heatmap showing the expression levels of cytokine genes. **m** Intracellular cytokine staining of As-iNKT1, Au-iNKT1, and A-iNKT17 cells from WT mice stimulated with PMA/Ionomycin. Representative FACS plots and proportion of cytokine-positive cells among each adipose iNKT cell subpopulation (TNFα, IL-17A (n = 4), and IFNγ (n = 5)). **n** Representative FACS plot and proportion of eGFP[+] cells among As-iNKT1, Au-iNKT1, and A-iNKT17 cells from IL-4/GFP enhanced transcript (4Get) mice (n = 6 biologically independent mice). Connected dots represent paired cell populations in a single stromal vascular fraction (SVF) (**m** and **n**). Data are represented as mean ± SD. n.s., non-significant. One-way ANOVA (**a**, **b**, **f**, **h**, **j**, and **k**). Paired one-way ANOVA (**m** and **n**). Two-tailed unpaired Student's t test (**d** and **i**). Two-tailed paired Student's t test (**g**). As-iNKT1; Adipose specific iNKT1, Au-iNKT1; Adipose universal iNKT1.

It is well known that immune cells could differentiate into subsets upon pathological stimuli[39]. To test whether As-iNKT1 cells could differentiate into other subpopulations in obesity, we performed pseudotime analysis on three iNKT1 cell subpopulations: Au-iNKT1, As-iNKT1, and Ac-iNKT1 cells. As shown in Fig. 3f, As-iNKT1 cells appeared to be derived from Au-iNKT1 cells and could differentiate into Ac-iNKT1 cells. Clonotype overlap analysis further verified that Ac-iNKT1 cells would share an origin with As-iNKT1 cells (Fig. 3g, h). Further, A-Cycling iNKT1 cells were the proliferating subset of As-iNKT1 and Ac-iNKT1 cells, which might explain their numerical increase in obesity (Fig. 3g). To investigate the hierarchy between Au-iNKT1, As-iNKT1, and Ac-iNKT1 cells in vivo, Au-iNKT1 and As-iNKT1 cells were injected into the left and right fat pads of HFD-fed iNKT cell-deficient Jα18 KO mice[40], respectively (Fig. 3i). It is interesting to note that Au-iNKT1 cells differentiated into both As-iNKT1 and Ac-iNKT1 cells, whereas As-iNKT1 cells only differentiated into Ac-iNKT1 cells in obesity (Fig. 3j–l). Moreover, the differentiation of As-iNKT1 cells was accompanied by a decrease in the CDR3β diversity of Ac-iNKT1 cells (Fig. 3m and Supplementary Fig. 4i), indicating that certain As-iNKT1 cells would differentiate into Ac-iNKT1 cells via clonal expansion. Taken together, these data propose that As-iNKT1 cells could differentiate into Ac-iNKT1 cells upon exposure to obesogenic stimuli.

### Ac-iNKT1 cells kill hypertrophic and inflammatory adipocytes and recruit macrophages by secreting CCL5

Recently, we have shown that adipose iNKT cells upregulate FasL upon HFD and remove hypertrophic and pro-inflammatory adipocytes[18]. The elevated number and cytotoxic gene expression of Ac-iNKT1 cells in obesity (Figs. 3d, 4a, b) prompted us to examine whether Ac-iNKT1 cells would be a major iNKT cell subpopulation that could remove enlarged and inflamed adipocytes. To address this, as shown in Fig. 4c, the same number of adipose iNKT cell subpopulations was cocultured with palmitic acid (PA)- or oleic acid (OA)-overloaded 3T3-L1 adipocytes (ADs), which have been considered as hypertrophic with pro-inflammatory characteristics or hypertrophic with no inflammatory characteristics, respectively (Supplementary Fig. 5a–d)[18,41]. To overcome the technical issue of small number of adipose iNKT cells, we adopted in vivo expansion of adipose iNKT cells by using alpha-galactosylceramide (α-GC), a potent lipid antigen for iNKT cells. Gene expression profiles and cytokine production of each adipose iNKT cell subpopulation were largely unaltered by in vivo expansion (Supplementary Fig. 5e–l). We firstly examined the correlation between AD size and death with each iNKT cell subpopulation. PA- or OA-treated 3T3-L1 ADs were categorized into two groups based on their lipid droplet (LD) size. Large ADs were defined as a diameter of LD > 14 μm, the proportion of which was increased by free fatty acid overloading (Supplementary Fig. 5a–c). Interestingly, large ADs were selectively killed by Ac-iNKT1 cells, whereas small ADs were less prone to be removed by iNKT cells (Fig. 4d, e). Next, to test whether cytotoxic activity of Ac-iNKT1 cells would depend on inflammatory

characteristics of ADs, PA-treated and OA-treated ADs, which differ in their inflammatory characteristics, were cocultured with each iNKT cell subpopulation. As indicated in Fig. 4f and Supplementary Fig. 5m, Ac-iNKT1 cells exhibited the greatest cytotoxicity against PA-treated 3T3-L1 ADs, hypertrophic and pro-inflammatory ADs, whereas they did not show significant cytotoxicity toward OA-treated 3T3-L1 ADs, hypertrophic but not inflammatory ADs. Also, Ac-iNKT1 cell-induced PA-treated AD death was largely decreased by CD1d neutralization (Supplementary Fig. 5n). These data suggest that Ac-iNKT1 cells would selectively remove adipocytes with both hypertrophic and pro-inflammatory characteristics via a TCR/CD1d-dependent manner. To scrutinize the cytotoxicity of Ac-iNKT1 cells in vivo, we injected the same number of adipose iNKT cell subpopulations into the right fat pad of HFD-fed iNKT cell-deficient Jα18 KO mice (Fig. 4g). A higher number of crown-like structures (CLSs), a marker of dead adipocytes[42], were detected in Ac-iNKT1 cell-injected fat pads (Fig. 4h, i). Moreover, Ac-iNKT1 cells showed similar gene expression profiles to terminally differentiated CD8[+] effector T cells (TE) (Supplementary Fig. 5o, p)[43], indicating that Ac-iNKT1 cells would play key roles to kill enlarged and inflamed adipocytes in obesity.

Macrophages are recruited around dead adipocytes to engulf them and form CLS[42,44,45]. To test the possibility that Ac-iNKT1 cells could mediate macrophage recruitment around dead adipocytes after killing them, cytokines involved in macrophage recruitment were examined[46]. As shown in Fig. 4j, k, Ccl5, the most highly expressed cytokine among macrophage-recruiting cytokines, was exclusively expressed in Ac-iNKT1 cells. Ccr5, a receptor for Ccl5[47], is exclusively expressed in macrophages, monocytes, and NK cells (Fig. 4l)[48], suggesting that CCL5 could play a role in the clearance of dead adipocytes by recruiting macrophages. Consistently, we observed that CCL5 could mediate the recruitment of THP1 monocytes (Fig. 4m). Taken together, these data clearly suggest that Ac-iNKT1 cells could selectively remove hypertrophic and pro-inflammatory adipocytes and would recruit macrophages via CCL5 to clean up dead adipocytes in obesity.

### A-iNKT17 cells stimulate adipose stem cell proliferation by secreting amphiregulin

Adipocyte turnover involves not only adipocyte death but also the generation of new adipocytes[3]. In this regard, activation of adipose iNKT cells could stimulate the proliferation of ASCs in obese mice[18]. Despite these, it is unknown whether ASC proliferation after adipose iNKT cell activation would be attributable to factor(s) from dead adipocytes or the direct interaction between adipose iNKT cells and ASCs. To test the hypothesis that iNKT cell activation could directly stimulate ASC proliferation without stimulating adipocyte death, α-GC was injected into lean mice. α-GC increased the proportion of proliferating ASCs without stimulating apoptosis in adipose tissue (Fig. 5a and Supplementary Fig. 6a). α-GC-induced ASC proliferation was abolished in iNKT cell-depleted Jα18 KO mice (Supplementary Fig. 6b). Also, increased ASC proliferation by α-GC further increased the total

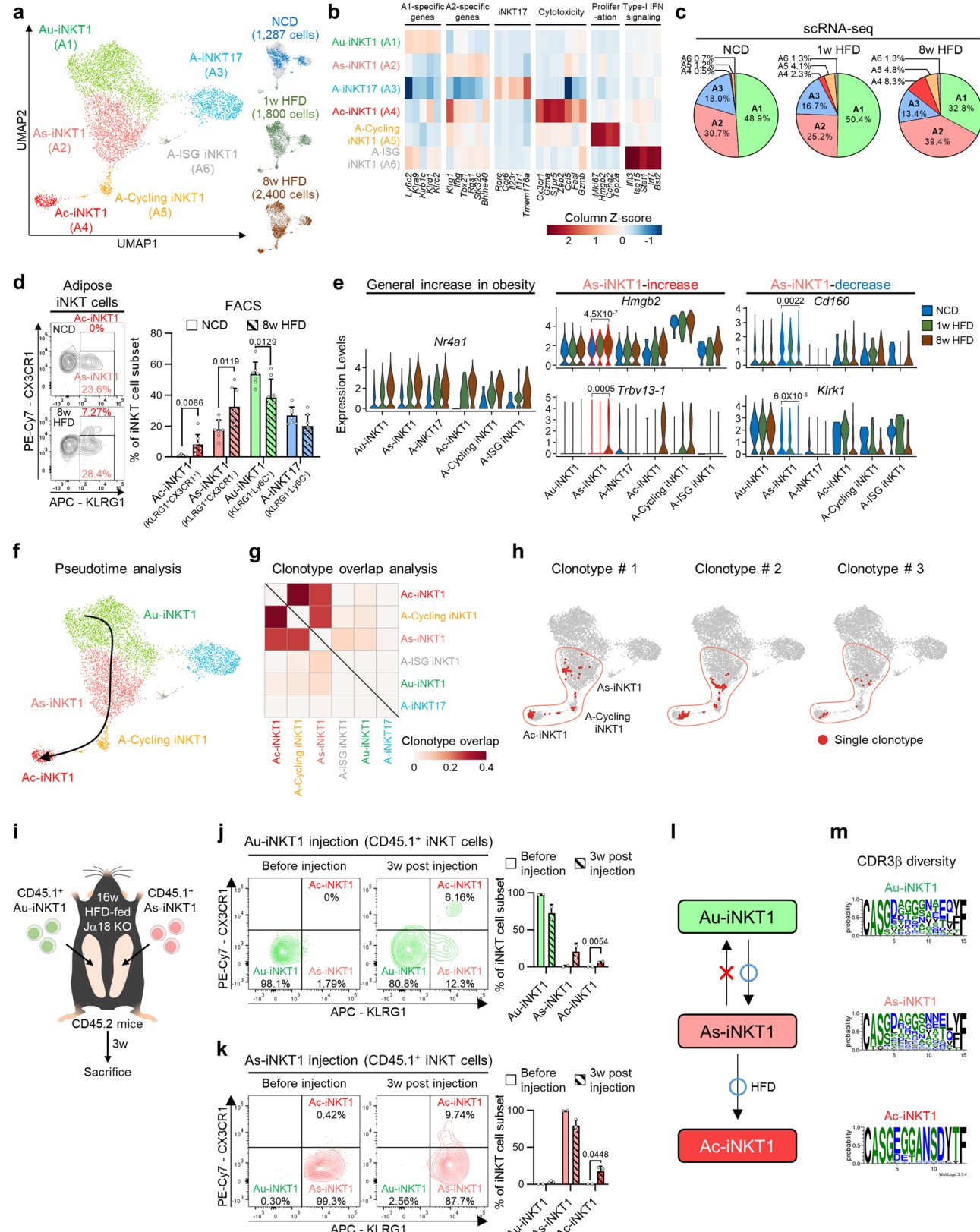

number of ASCs in WAT (Fig. 5b). These data suggest that adipose iNKT cells could directly stimulate ASC proliferation upon activation.

To understand which adipose iNKT cell subpopulations would be involved in ASC proliferation, we used in vivo approaches. As the ratio of A-iNKT17 cells was significantly increased by α-GC injection (Fig. 5c), we attempted to verify the role of A-iNKT17 cells in ASC proliferation.

When A-iNKT17 cells were injected into the fat pads of HFD-fed Jα18 KO mice, the ratio of proliferating ASCs was increased by injection of A-iNKT17 cells, whereas those of other adipose iNKT cell subpopulations were not (Fig. 5d and Supplementary Fig. 6c, d). Then, to identify which factor(s) from A-iNKT17 cells could stimulate ASC proliferation, we examined gene expression profiles in adipose iNKT cell

**Fig. 3 | In obesity, As-iNKT1 cells give rise to Ac-iNKT1 cells. a** Unsupervised clustering of TCRβ^int/CD1d.PBS57 tetramer⁺ iNKT cells from WAT of NCD-, 1-week, or 8-week HFD-fed mice. 1,287 cells from NCD-, 1,800 cells from 1-week, and 2,400 cells from 8-week HFD-fed mice on a UMAP plot. **b** Heatmap showing the expression levels of subpopulation marker genes. **c** Proportion of each adipose iNKT cell subpopulation in scRNA-seq data. **d** Representative FACS plots and proportion of each subpopulation among total adipose iNKT cells in NCD- ($n = 7$) or 8-week HFD-fed mice ($n = 8$). **e** Gene expression levels of *Nr4a1*, *Hmgb2*, *Trbv13-1*, *Cd160*, and *Klrk1* in adipose iNKT cell subpopulations under each condition. p-values were adjusted for multiple comparisons using Bonferroni correction. **f** In silico pseudotime analysis of adipose iNKT1 cells. **g** Clonotype overlap analysis of adipose iNKT cell subpopulations. **h** Representative clones showing clonotype overlapping between As-iNKT1, Ac-iNKT1, and A-Cycling iNKT1 cells. **i** Experimental scheme for adoptive transfer of CD45.1⁺ Au-iNKT1 and As-iNKT1 cells. iNKT cells were sorted from CD45.1 mice 1-week after α-GC injection and injected into each WAT fat pad of 16-week HFD-fed CD45.2 Jα18 KO mice. **j, k** Representative FACS plots and composition of injected CD45.1⁺ donor iNKT cells in recipient mice after 3 weeks (Before injection ($n = 2$), Au-iNKT1 post injection ($n = 2$), and As-iNKT1 post injection ($n = 3$)). **l** Schematic diagram of iNKT1 differentiation process in adipose tissue. **m** CDR3β amino acid sequences of most prevalent CDR3β length of each adipose iNKT1 cell subpopulation. Data were collected from 8-week-HFD condition. Data are represented as mean ± SD. n.s., non-significant. Two-tailed unpaired Student's t test (**d, e, j**, and **k**). Au-iNKT1; Adipose universal iNKT1, As-iNKT1; Adipose specific iNKT1. Ac-iNKT1; Adipose cytotoxic iNKT1.

subpopulations. It has been reported that immune cells actively crosstalk with epithelial or mesenchymal cells during tissue regeneration via growth factors or cytokines[49–52]. Interestingly, among previously reported immune cell-derived growth factors[50], amphiregulin (*Areg*), reported to stimulate satellite cell differentiation during muscle repair[53], was exclusively expressed in A-iNKT17 cells, and its expression level was gradually increased upon HFD (Fig. 5e). Also, mRNA level of *Areg* appeared to be abundant in A-iNKT17 cells and was downregulated in iNKT cell-depleted mice upon HFD feeding (Fig. 5f, g). To examine whether AREG could stimulate ASC proliferation, lean mice were injected with AREG. As indicated in Fig. 5h, AREG potentiated the proportion of proliferating ASCs in a dose-dependent manner. Also, α-GC-induced ASC proliferation was decreased by the inhibition of epidermal growth factor receptor (EGFR), a receptor for AREG (Supplementary Fig. 6e)[54]. Next, to investigate whether ASC proliferation stimulated by AREG would indeed lead to adipogenesis, we adopted adipocyte lineage-tracing mice (adiponectin^rtTA; TRE-Cre; rosa26-loxp-stop-loxp-YFP). Pre-existing adipocytes were marked with YFP by doxycycline, whereas newly formed adipocytes after vehicle or AREG injection were not labeled by YFP (Fig. 5i). The proportion of perilipin-positive and YFP-negative new adipocytes was increased by AREG injection (Fig. 5j, k), implying that AREG-driven ASC proliferation would increase adipogenic potential. In addition, it seemed that AREG-injected WAT showed a smaller adipocyte size (Fig. 5l), which was observed in adipose tissue with activated adipogenesis[55]. Given that the proportion of iNKT17 cells was higher in WAT compared to thymus, spleen, and liver, it seems that A-iNKT17-AREG axis appeared to play a substantial role in the regulation of ASC proliferation (Supplementary Fig. 6f). Collectively, these data suggest that A-iNKT17 cell-derived AREG could be one of the key mediators to stimulate ASC proliferation and adipogenesis in adipose tissue homeostasis.

## Discussion

Adipose tissue-resident iNKT cells play important roles in adipose tissue homeostasis by controlling adipocyte quantity, quality, and inflammation[16–18,32]. However, it is still unclear how iNKT cells acquire and mediate numerous functions in adipose tissue. In this study, we identified unique subpopulations of iNKT cells in WAT and the underlying mechanisms by which adipose iNKT cells control adipocyte turnover and inflammation in obesity. In obese WAT, Ac-iNKT1 cells differentiated from As-iNKT1 cells selectively removed bad adipocytes and recruited macrophages to clear dead adipocytes by secreting CCL5. In addition, we found that A-iNKT17 cells stimulated ASC proliferation and adipogenesis through AREG secretion. Together, these data propose that the unique adipose iNKT cell subpopulations orchestrate adipocyte turnover through dynamic interactions with other cell types such as adipocytes, adipose immune cells, and adipose stem cells (Fig. 6).

Adipose iNKT cells have tissue-specific characteristics[25]. However, the acquisition and role of these characteristics in adipocyte turnover have not been adequately explored. Here, we discovered adipose tissue-specific iNKT cell subpopulations and their differentiation

pathways by performing scRNA-seq analysis of thymic and adipose iNKT cells. Projection analysis of thymic, splenic, hepatic, and lymph node iNKT cells uncovered that KLRG1⁺ As-iNKT1 cells barely overlapped with iNKT cells in other organs. Furthermore, we found that the generation of As-iNKT1 cells is promoted by adipose tissue microenvironment in a CD1d-independent manner. Previously, KLRG1⁺ iNKT cells have been reported in peripheral organs such as the liver, lung, and spleen after injection of α-GC[56,57]. KLRG1⁺ iNKT cells also exhibit Th1-polarized phenotypes in their cytokine profiles, cytotoxicity, and transcription factor expression[56], similar to those of As-iNKT1 cells. Although these findings imply that KLRG1⁺ iNKT cells would be generated by α-GC, a strong iNKT cell stimulator, in various organs, we found that in lean adipose tissue, KLRG1⁺ iNKT1 cells were produced in adipose tissue microenvironment, independent of CD1d-mediated activation processes. In WAT, three major iNKT1 cell subpopulations appeared to be hierarchical in the order of Au-iNKT1, As-iNKT1, and obesity-induced Ac-iNKT1 cells. Considering that the transitions from Au-iNKT1 to As-iNKT1 and Ac-iNKT1 cells were promoted by HFD (Fig. 3i–l, and Supplementary Fig. 3d–f), it is feasible to speculate that obesity-induced factors would facilitate Au-iNKT1 to As-iNKT1 and Ac-iNKT1 cell transition. In addition, adipocyte-derived secretory factors could upregulate some As-iNKT1 characteristic gene expressions. Recently, it has been suggested that distinct characteristics of adipose tissue-resident immune cells would be determined by adipose tissue microenvironment with high contents of lipid metabolites, adipokines, and sex hormones[26,58,59]. Future studies are needed to investigate which factor(s) from the adipose tissue microenvironment could induce adipose-specific iNKT cell generation. Together, it is likely that adipose iNKT cells would undergo a distinct differentiation process in the adipose tissue microenvironment and be endowed with the ability to protect adipose tissue, probably, through differentiation into a cytotoxic subpopulation.

Adipocyte death is frequently observed in obesity and is closely related to adipose tissue inflammation and whole-body energy metabolism[10,42,60]. In obesity, adipocyte death occurs due to various stimuli, such as mechanical stress[61], hypoxia[62,63], hypertrophy[42], and adipose iNKT cells[18]. Components from dead adipocytes, such as lipids or cholesterol, should be sequestered and removed from the interstitium to minimize their harmful effects on neighboring cells[42]. Accordingly, it is important to temporally coordinate apoptosis induction and elimination of apoptotic cells. A very recent paper showed that macrophages are recruited around laser irradiation-induced dead adipocytes and form CLS within 72 hours under ex vivo conditions[64]. Nevertheless, it remains unknown how apoptosis induction and efficient clearance are elaborately controlled in obese adipose tissue. Here, we suggest that Ac-iNKT1 cells differentiated from As-iNKT1 cells would selectively get rid of hypertrophic and pro-inflammatory adipocytes and recruit phagocytic macrophages to clear them. Ac-iNKT1 cells exclusively expressed *Zeb2*, *Cx3cr1*, and *Gzma*, which are highly expressed in CD8⁺ effector T cells[43], implying that Ac-iNKT1 cells would be potent killer cells. Consistently, coculture experiments and adoptive transfer of iNKT cell subpopulations

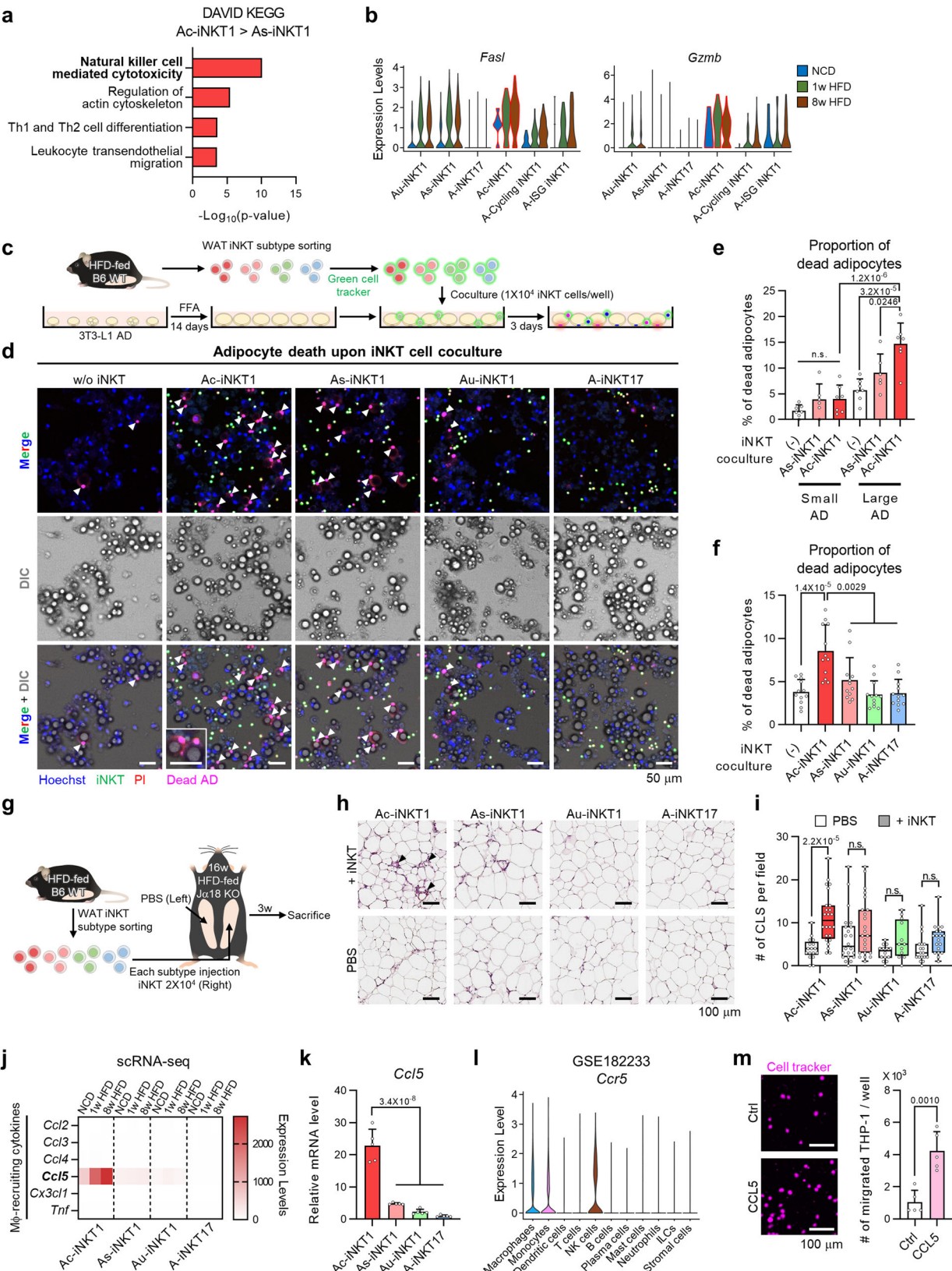

**b** *Fasl* and *Gzmb* expression levels across iNKT subtypes: NCD, 1w HFD, 8w HFD.

**c** Schematic: HFD-fed B6 WT → WAT iNKT subtype sorting → Green cell tracker → Coculture (1X10⁴ iNKT cells/well). 3T3-L1 AD, FFA 14 days, 3 days.

**d** Adipocyte death upon iNKT cell coculture. Merge / DIC / Merge + DIC. w/o iNKT, Ac-iNKT1, As-iNKT1, Au-iNKT1, A-iNKT17. Hoechst, iNKT, PI, Dead AD. 50 μm.

**e** Proportion of dead adipocytes.

**f** Proportion of dead adipocytes.

**g** Schematic: HFD-fed B6 WT → WAT iNKT subtype sorting → Each subtype injection iNKT 2X10⁴ (Right); PBS (Left); 16w HFD-fed Jα18 KO; 3w → Sacrifice.

**h** Ac-iNKT1, As-iNKT1, Au-iNKT1, A-iNKT17 (+ iNKT / PBS). 100 μm.

**i** # of CLS per field. PBS / + iNKT.

**j** scRNA-seq. Mφ-recruiting cytokines: *Ccl2*, *Ccl3*, *Ccl4*, *Ccl5*, *Cx3cl1*, *Tnf*.

**k** *Ccl5* relative mRNA level.

**l** GSE182233 *Ccr5*.

**m** Cell tracker. Ctrl / CCL5. # of migrated THP-1 / well. 100 μm.

indicated that Ac-iNKT1 cells facilitated the death of unhealthy adipocytes. Given that target specificity of iNKT cells depends on the type of Vβ and CDR3β sequence[65–67], it appears that Ac-iNKT1 cells with highly invariable TCR chains might kill certain adipocytes bearing specific lipid antigens. Moreover, we found that Ac-iNKT1 cells highly expressed chemoattractant, *Ccl5*, whose expression in Ac-iNKT1 cells

gradually increased upon HFD. CCL5 is known for macrophage recruitment and survival in obese adipose tissue[68]. Recent scRNA-seq analyses have shown that cytotoxic immune cells such as NK cells, CD8+ T cells, and NKT cells highly express *Ccl5* in adipose tissue[48,69], implying that cytotoxic immune cells would contribute to effective clearance of dead cells via macrophage recruitment. Taken together,

**Fig. 4 | Ac-iNKT1 cells kill hypertrophic and inflammatory adipocytes and recruit macrophages by secreting CCL5. a** KEGG pathway analysis of Ac-iNKT1 high-differentially expressed genes (DEGs) compared to As-iNKT1 cells ($P < 0.05$). **b** Expression levels of cytotoxic marker genes in adipose iNKT cell subpopulations. **c** Experimental design for coculture of each iNKT cell subpopulation with hypertrophic adipocytes. **d** Representative images of coculture between PA-treated 3T3-L1 adipocytes and iNKT cell subpopulations. Arrows; PI⁺ dead adipocytes. Scale bars, 50 μm. **e, f** Proportion of PI⁺ adipocytes among large or small adipocytes ((-) ($n = 7$), As-iNKT1 ($n = 6$), and Ac-iNKT1 ($n = 7$)) (**e**) and among total adipocytes ($n = 12$ biologically independent cells) (**f**). Cells in (**d**) were counted by using microscope (**e** and **f**). **g** Experimental scheme for injection of each iNKT cell subpopulation. iNKT cells were sorted from 8-week HFD-fed WT male mice 1-week after α-GC injection and injected into HFD-fed Jα18 KO mice. **h** Representative histological images of PBS or iNKT cell subpopulation-injected WAT fat pads. Arrows; CLS. Scale bars, 100 μm. **i** Quantification of the number of CLS (Ac-iNKT1/PBS ($n = 17$), Ac-

iNKT1/+iNKT ($n = 24$), As-iNKT1/PBS ($n = 22$), As-iNKT1/+iNKT ($n = 23$), Au-iNKT1/PBS ($n = 14$), Au-iNKT1/+iNKT ($n = 12$), A-iNKT17/PBS ($n = 18$), and A-iNKT17/+iNKT ($n = 19$)). **j** Heatmap showing the expression levels of macrophage-recruiting cytokines in adipose iNKT cell subpopulations. **k** *Ccl5* mRNA level in in vivo-expanded iNKT cell subpopulations sorted from 10-week-old male mice ($n = 5$ biologically independent samples). **l** *Ccr5* expression level in immune cells from WAT (GSE182233). **m** Left: representative images of monocyte infiltration with or without CCL5 peptides. Right: quantification of the number of infiltrated THP1 cells ($n = 5$ biologically independent samples). Scale bars, 100 μm. Data are represented as mean ± SD except (**i**) represented as box and whiskers plot. In (**i**), the lower, central, and upper line in each box represents the first (Q1), the second (median), and the third quartile (Q3), respectively. The whiskers extend from the box to the minimum and maximum observations, respectively. n.s., non-significant. One-way ANOVA (**f** and **k**). Two-way ANOVA (**e** and **i**). Two-tailed unpaired Student's t test (**m**). One-tailed Fisher's exact test (**a**).

these data suggest that Ac-iNKT1 cells play crucial roles in the control of adipocyte turnover, particularly in obesity.

Immune cells are one of the key factors to determine the properties and fate of stem cells[70,71]. For instance, in muscle and intestine, several immune cells such as macrophages, innate lymphoid cells (ILCs), T helper cells, and Regulatory T (Treg) cells mediate stem cell proliferation and differentiation through secreting epidermal growth factor (EGF) family[53,72]. EGFs activate cellular proliferation, differentiation, and survival by binding to their cognate receptors[73]. Despite these findings, the importance of crosstalk between immune cells and stem cells in adipose tissue remains largely unknown. In this study, we found that A-iNKT17 cells promoted the proliferation of ASCs via secretion of AREG, a member of the EGF family, which was followed by adipogenesis. Our previous reports[14,18] and current studies have shown that activation of adipose iNKT cells would be accompanied by ASC proliferation. In adipose tissue, it has been reported that Th2 CD4⁺ T cells, ILC2, and Treg cells express AREG[74,75]. In this study, we have identified that A-iNKT17 could also secrete AREG. Although we assumed that AREG-EGFR signaling in WAT would be important for ASC proliferation, it is not fully understood whether A-iNKT17 cells directly secrete AREG or indirectly stimulate other cell types (e.g. Treg cells or ILC2) to secrete AREG. Future studies are required to determine the complex interactions between immune cells and ASCs in adipose tissue. In addition, given that A-iNKT17 cells highly expressed IL-4, it is likely that A-iNKT17 cells would be involved in the resolution of adipose tissue inflammation. Together, current data suggest that A-iNKT17 cells would contribute to healthy adipose tissue remodeling, potentially, by regulating ASC proliferation, adipocyte differentiation, and inflammation.

There have been controversies in the roles of adipose iNKT cells in adipose tissue inflammation. We and others have suggested that adipose iNKT cells exert protective roles with anti-inflammatory cytokines such as IL-4 and IL-10[16,17,19,32], whereas other groups have reported that they exert pathogenic roles with pro-inflammatory cytokines such as IFNγ and TNFα[76,77]. In this study, we found that adipose iNKT cells consist of both pro-inflammatory and anti-inflammatory subpopulations and both of them coordinate adipocyte turnover by regulating adipocyte death and ASC proliferation, respectively. These functionally discrete subpopulations might, at least partly, explain the discrepancies regarding the roles of adipose iNKT cells in a context-dependent manner.

In conclusion, we characterized adipose iNKT cells at a single-cell resolution and identified the mechanisms and physiological significance of iNKT cell-mediated adipocyte turnover. Our comprehensive analyses using scRNA-seq and experimental approaches provide substantial insights into roles of adipose immune cells in the control of adipocyte quantity and quality. Collectively, our data suggest that dynamic cellular crosstalk between iNKT cells, macrophages, and (pre) adipocytes in adipose tissue is crucial for the maintenance of adipose tissue homeostasis.

## Methods
### Animals and treatments
2 - 24-week-old C57BL/6N male mice, 10-week-old C57BL/6N female mice, and 10-week-old BALB/c (000651, The Jackson Laboratory) male mice were purchased from JA BIO Incorporation (Suwon-si, Gyeonggi-do, Republic of Korea). CD1d KO[78] and Jα18 KO[40] mice were gifted by Doo Hyun Chung (Seoul National University College of Medicine, Seoul, Republic of Korea). CD45.1 (002014, The Jackson Laboratory) and 4Get mice (004190, The Jackson Laboratory) were gifted by Ro Hyun Seong (Seoul National University, Seoul, Republic of Korea). 8 - 14-week-old adipocyte lineage-tracing mice (adiponectinʳᵗᵀᴬ (033448, The Jackson Laboratory); TRE-Cre (006234, The Jackson Laboratory); rosa26-loxp-stop-loxp-YFP (006148, The Jackson Laboratory)) (CL57BL/6J) were obtained from the UNIST. The mice were housed in a specific pathogen-free, temperature- (22 °C) and humidity- (50%) controlled animal facility under a 12-h/12-h light/dark cycle. High-fat diet (HFD) feeding experiments were performed using mice older than 8 weeks of age and fed a diet consisting of 60% calories from fat (D12492, Research Diets). Adipocyte lineage tracing mice were fed a doxycycline (600 mg/kg)-containing NCD for 2 weeks from 8 weeks of age. To expand adipose iNKT cells in vivo, mice were intraperitoneally injected with α-galactosylceramide (α-GC) (1 μg/mouse, AG-CN2-0013, AdipoGen) and sacrificed after 1 week. Amphiregulin (1 μg/mouse, 989-AR, R&D Systems) or Gefitinib (20 mg/kg, S1025, Selleckchem) was intraperitoneally injected once a day for three consecutive days and mice were sacrificed on the next day. The animal studies were approved by the Institutional Animal Care and Use Committee of the Seoul National University.

### Human participants
Human omental adipose tissue samples were obtained during weight reduction laparoscopic bypass surgery at the Metabolic Surgery Center in Seoul National University Bundang Hospital (SNUBH). This study was conducted in accordance with the Declaration of Helsinki and was approved by the Ethics Committee of the SNUBH (IRB No. B-1801445301 & B-1812513302). All participants provided their written informed consent. All participants were female, aged 30-50 years, and BMI between 35-50 kg/m². Tissue samples consisted of 25-mg tissue blocks. The processing of the tissues was initiated within 3 hours after removal from the patients without snap-frozen or cryo-preservation.

### scRNA-seq library preparation
iNKT cells (TCRβⁱⁿᵗ/CD1d.PBS57 tetramer⁺) were sorted from stromal vascular fraction (SVF) of epididymal white adipose tissue and cell suspension from thymus by flow cytometry (Supplementary Fig. 1a). Tissue processing procedures are described below in adipose tissue fractionation and lymphocyte preparation sections. The sequencing library was generated using Chromium Single Cell 5′ Library & Gel Bead Kit (PN-1000014, 10X Genomics), Chromium Single Cell A Chip kit (PN-

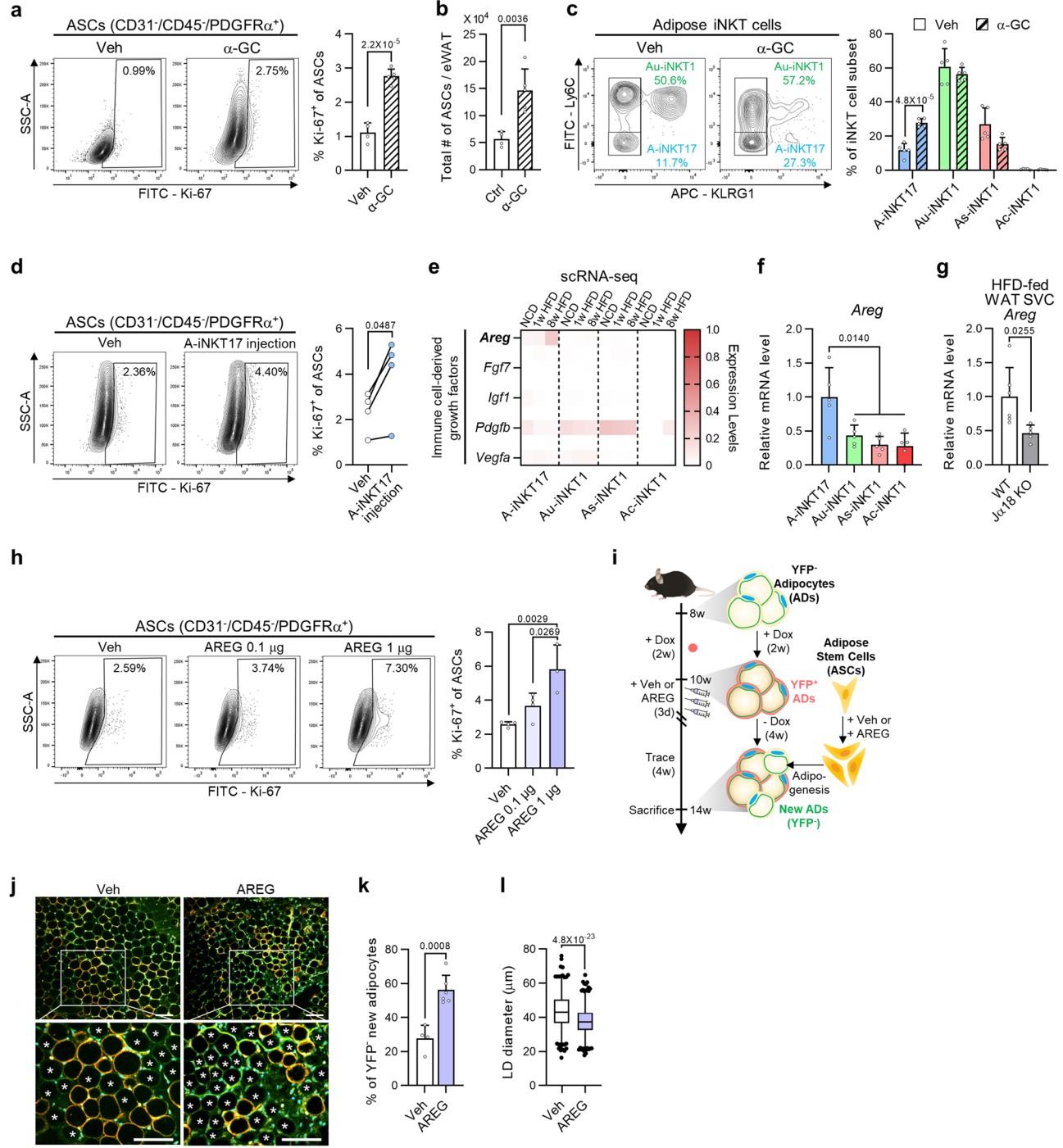

**Fig. 5 | A-iNKT17 cells stimulate adipose stem cell proliferation by secreting amphiregulin. a** Representative FACS plots and proportion of Ki-67$^+$ among ASCs (CD31$^-$/CD45$^-$/PDGFRα$^+$) from WT WAT 4-days after vehicle ($n = 4$) or α-GC injection ($n = 5$). **b** The total number of ASCs from WT WAT 1-week after vehicle ($n = 4$) or α-GC injection ($n = 5$). **c** Representative FACS plots and proportion of each adipose iNKT cell subpopulation among total adipose iNKT cells 4-days after vehicle or α-GC injection ($n = 5$ biologically independent mice). **d** Representative FACS plots and proportion of Ki-67$^+$ among ASCs from WAT of 12-week HFD-fed Jα18 KO mice with or without A-iNKT17 cell injection ($n = 4$ biologically independent mice). Mice were sacrificed 4 days after injection. **e** Heatmap showing the expression levels of growth factors in adipose iNKT cell subpopulations. **f** *Areg* mRNA level in in vivo-expanded iNKT cell subpopulations sorted from 10-week-old male mice ($n = 5$ biologically independent samples). **g** *Areg* mRNA level in WAT SVFs from 8-week HFD-fed WT ($n = 6$) and Jα18 KO mice ($n = 5$). **h** Representative FACS plots and

proportion of Ki-67$^+$ among ASCs from WT WAT. Mice were injected with vehicle, AREG 0.1 μg ($n = 4$), or AREG 1 μg ($n = 3$). **i** Experimental scheme using C57BL/6J adipocyte lineage-tracing male mice. **j** Representative microscopic images of vehicle or AREG injected WAT 4 weeks after injection. Asterisks indicate perilipin$^+$/YFP$^-$ new adipocytes. Scale bars, 100 μm. **k** Proportion of YFP$^-$ new adipocytes among total adipocytes from vehicle ($n = 4$) or AREG injected mice ($n = 6$). **l** Quantification of lipid droplet (LD) size of vehicle ($n = 489$) or AREG injected WAT ($n = 527$) in (**j**). Data are represented as mean ± SD except (**l**) represented as box and whiskers plot. In (**l**), the lower, central, and upper line represents the first (Q1), the second (median), and the third quartile (Q3), respectively. The whiskers extend from the box to the 2.5 and 97.5 percentile, respectively. n.s., non-significant. One-way ANOVA (**f** and **h**). Two-tailed unpaired Student's t test (**a**–**c**, **g**, **k**, and **l**). Two-tailed paired Student's t test (**d**).

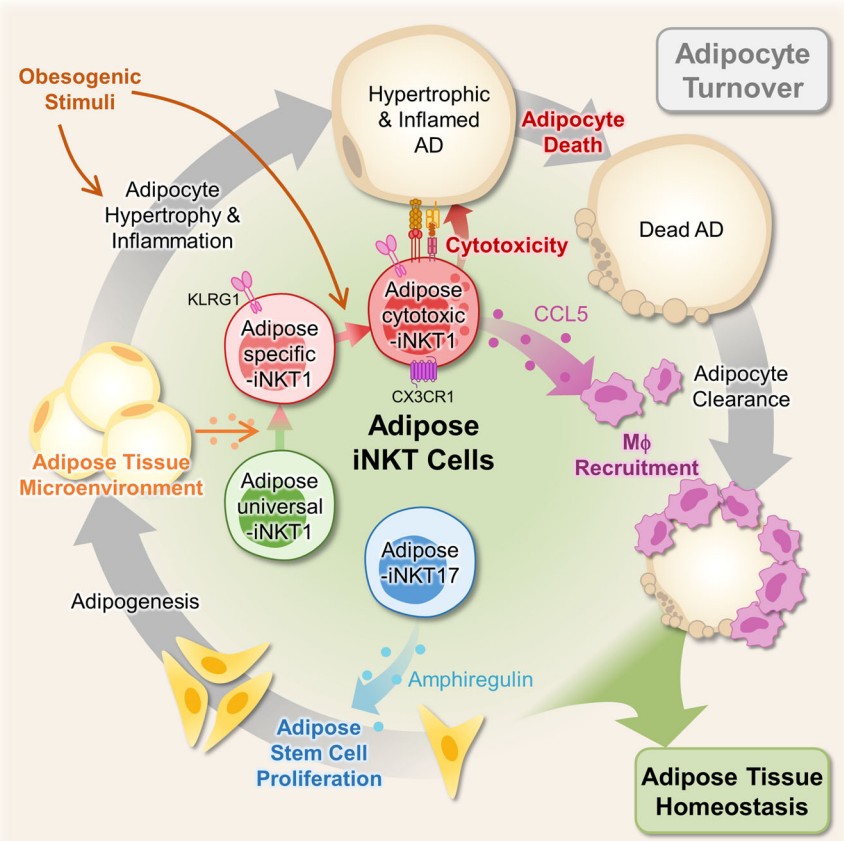

**Fig. 6 | Proposed Model.** Adipose iNKT cells are composed of adipose-specific (As)-iNKT1 cells which possess distinct transcriptome profiles including KLRG1. As-iNKT1 cells are differentiated from adipose-universal (Au)-iNKT1 cells under adipose tissue microenvironment. In obesity, some As-iNKT1 cells differentiate into adipose-cytotoxic (Ac)-iNKT1 cells. Ac-iNKT1 cells are highly cytotoxic and they selectively kill large and inflamed adipocytes. Also, they recruit macrophages around dead adipocytes by secreting CCL5. Adipose iNKT17 cells stimulate the proliferation of adipose stem cells by amphiregulin secretion, helping adipose tissue regeneration.

1000009, 10X Genomics), and Chromium i7 Multiplex Kit (PN-120262, 10X Genomics). Single-cell suspension was loaded on the Single Cell A Chip (10X Genomics) and run in the Chromium Controller to generate gel bead-in-emulsions (GEMs) aiming to capture 4000 cells per channel. Following reverse transcription was performed using C1000 thermocycler (Bio-Rad). Subsequent cDNA purification and library generation were performed according to the manufacturer's instructions provided. To generate single-cell TCR sequencing library, 1/22.5 of total cDNA was used for V(D)J target enrichment PCR with mouse T cell specific primer set (PN-1000071, 10X Genomics). The quality of libraries was confirmed using a Bioanalyzer High Sensitivity DNA kit (5067-4626, Agilent). Libraries were sequenced on an Illumina HiSeqX10 (paired-end 100 bp reads) aiming at an average of 50,000 read pairs (transcriptome libraries) or 5000 read pairs (TCR libraries) per cell.

### scRNA-seq data analysis

Raw reads of sorted iNKT cells were mapped to a mouse reference genome (GRCm38) using the Cell Ranger software (v3.1.0) and the GRCm38.99 GTF file. A gene-by-cell count matrix for each sample was generated with default parameters, except for expected cells = 4000, 3000 (Adipose 1wHFD), or 2500 (Adipose 8wHFD). Empty droplets were excluded using the emptyDrops function of the DropletUtils (v1.6.1) R package[79] with a false discovery rate (FDR) < 0.05. To filter out poor quality cells, cells with less than 1000 unique molecular identifiers (UMIs) and higher than 10% of UMIs mapped to mitochondrial genes were excluded using the calculateQCMetrics function of

the scater (v1.14.0) R package[80]. To remove cell-specific biases, cell-specific size factors were calculated using the computeSumFactors function of the scran (v1.14.6) R package[81]. The aggregated UMI count matrix were divided by cell-specific size factors and log2 transformed by adding a pseudocount of highly variable genes (HVGs) which were defined as genes with FDR < 0.05 for biological variability using the modelGeneVar function of the scran package. All cells across the samples were clustered into 21 clusters using the FindClusters function of the Seurat (v3.1.5) R package[82] on the first 15 principal components (PCs) of HVGs with resolution = 0.8. Clusters annotated as non-T and CD8+ T cell clusters were removed. After removing non-iNKT cell clusters, cells were grouped into 15 clusters using the same method described above with the first 20 PCs. Cells annotated as non-T, gamma delta T, innate lymphoid cell, CD8+ T cell clusters were excluded. Cells were regrouped into 13 clusters using the same method as described above with the first 15 PCs. Cells in the normal chow diet (NCD) and only adipose iNKT cells of NCD were grouped into 10 and 8 clusters, respectively, and visualized in the UMAP plot using the same method, with the first 15 PCs of 1000 HVGs. Cells in adipose iNKT cells across diet conditions were clustered into 9 clusters and visualized in the UMAP plot using same method except for the first 15 PCs of HVGs with FDR < 0.05 for biological variability.

To identify cell type-specific marker genes, the FindAllMarkers function of the Seurat package was used. Cell type signature score was calculated by the AddModuleScore function of the Seurat package. For gene ontology analysis, a database for annotation, visualization and integrated discovery (DAVID)[83,84] was used. To project thymic, splenic,

hepatic, and lymph node iNKT cells onto NCD adipose iNKT cells, for every non-adipose iNKT cell, k-Nearest Neighbors (k-NNs, k = 5) were inferred from NCD adipose iNKT cells with respect to the Pearson correlation coefficients of normalized expression data of HVGs for NCD adipose iNKT cells by using the knn.index.dist function of the KernelKnn (v1.0.8) R package[85]. To obtain the projection coordinate of non-adipose iNKT cells in UMAP plot, two-dimensional coordinates of 5-NNs in the UMAP plot of NCD adipose iNKT cells were averaged. Pseudotime analysis was performed for adipose iNKT cells using the slingshot (v1.4.0) R package[86] based on the UMAP coordinates. To obtain a differentiation trajectory, the getLineages function of the slingshot R package was used with setting a start cluster as Au-iNKT1, and an ending cluster as Ac-iNKT1.

## TCR repertoire analysis

Raw reads from paired V(D)J sequencing were processed using the cellranger vdj function of the Cell Ranger software (v3.1.0). To construct V(D)J segment-based reference, cellranger mkvdjref function of the Cell Ranger was used with mouse V(D)J segment sequences from international ImMunoGeneTics information system (IMGT). For further analysis, we used contigs assigned as productive and high-confidence. Cells sharing the same V/J composition and identical CDR3 sequences both in heavy and light chains are regarded as the same clonotype. To calculate normalized diversity index, the ComputeShannonIndex function of the Stcr R package[87] was processed. CDR3 logo sequences were visualized by using WebLogo (v3.7.4).

## Adipose tissue fractionation

Epididymal white adipose tissue (WAT) were obtained from age-matched NCD-fed and 1- or 8-week HFD-fed mice. Adipose tissues were fractionated as described previously[88], with minor modifications. Briefly, adipose tissues were minced and digested with collagenase buffer [0.1 M HEPES (Sigma-Aldrich), 0.125 M NaCl, 5 mM KCl, 1.3 mM CaCl$_2$, 5 mM glucose, 1.5% (w/v) bovine serum albumin (BSA) (A0100-010, GenDEPOT), and 0.1% (w/v) collagenase I (49A18993, Worthington)] in a shaking incubator at 37 °C for 30 min. After centrifugation at 450 × $g$, 4°C for 3 min, the pelleted SVF was collected. The SVF was incubated in red blood cell (RBC) lysis buffer (17 mM Tris, pH 7.65, and 0.16 M NH$_4$Cl) for 3 min. Then, the SVFs were washed with phosphate-buffered saline (PBS) several times, passed through a 100-μm filter (93100, SPL), and collected by centrifugation at 450 × $g$ for 3 min.

## Isolation of SVFs from human adipose tissue

Human adipose tissues were rinsed in PBS twice, manually minced, and digested with collagenase buffer [0.1 M HEPES, 0.125 M NaCl, 5 mM KCl, 1.3 mM CaCl$_2$, 5 mM glucose, 1.5% (w/v) BSA, and 0.1% (w/v) collagenase I] in a shaking water bath at 37 °C for 30–60 min. The subsequent steps were the same as those for preparing mouse adipose tissue fractionation.

## Lymphocyte preparation from liver, spleen, thymus, and blood

Liver mononuclear cells (MNCs) were isolated using Percoll gradient as previously described[89]. Briefly, liver was gently passed through a nylon mesh and suspended in PBS. The cell suspension was centrifuged at 450 × $g$, 4°C for 3 min. The obtained cell pellet was resuspended in 40% Percoll (17-0891-01, Cytiva). Resuspended cell solution was carefully layered onto 70% Percoll and centrifuged at 800 × $g$, room temperature (RT) for 20 min with no break. MNCs were isolated from the middle layer. Then, the MNCs were washed with PBS several times, passed through a 100-μm filter (93100, SPL), and collected by centrifugation at 450 × $g$ for 3 min.

To isolate thymic or splenic lymphocytes, the thymus or spleen was mechanically disrupted in between two glass slides. Obtained cell suspension was centrifuged at 450 × $g$, 4°C for 3 min. The cell pellet was incubated in RBC lysis buffer for 3 min, washed with PBS several

times, passed through a 100-μm filter, and collected by centrifugation at 450 × $g$ for 3 min.

To isolate blood MNCs, blood samples were collected in a Greiner Leucosep tube (GN163290, Sigma Aldrich) pre-equilibrated with 3 mL of NycoPrep 1.077 (1114550, Axis-Shield PoC AS). After centrifugation at 450 × $g$, RT for 10 min, the middle layer was carefully isolated, washed with PBS several times, passed through a 100-μm filter, and collected by centrifugation at 450 × $g$ for 3 min.

## Fluorescence-activated cell sorting (FACS)

FACS was carried out as previously described[88], with minor modifications. Single-cell suspensions were incubated in Fc-receptor blocking antibody (1:300, 101302, BioLegend) at RT for 15 min prior to surface antigen staining. Then, the cells were stained with anti-TCR β chain (1:300, 109220, BioLegend), CD1d.PBS57 tetramer (1:300, NIH Tetramer Core Facility), anti-KLRG1 (1:300, 138412 or 138418, BioLegend), anti-Ly6C (1:300, 128006 or 128012, BioLegend), anti-CX3CR1 (1:300, 149020 or 149016, BioLegend), anti-CD45.1 (1:300, 110722, BioLegend), anti-human CD3 (1:300, 300317, BioLegend), or anti-human TCR Vα24-Jα18 (1:300, 342903, BioLegend) at 4°C for 30 min. To detect intracellular proteins, the SVFs were fixed, permeabilized with Foxp3 / Transcription Factor Staining Buffer Set (00-5523-00, Thermo Fisher), and stained with anti-T-bet (1:100, 644810, BioLegend), anti-PLZF (1:100, 53-9320-80, Thermo Fisher), anti-RORγt (1:100, 25-6981-82, Thermo Fisher), or anti-Ki-67 (1:300, 151212, BioLegend) for 30 min. To detect cytokines, fixed and permeabilized cells were stained with anti-IFNγ (1:100, 557649, BD), anti-TNFα (1:100, 506324, BioLegend), or anti-IL-17A (1:100, 506922, BioLegend) for 30 min. Adipose stem cells were identified by anti-CD45 (1:300, 103132, BioLegend), anti-CD31 (1:300, 102410, BioLegend), and anti-PDGFRα (1:300, 562776, BD) among SVFs and their proliferation was assessed by anti-Ki-67 (1:200, 151212, BioLegend). The cells were analyzed or sorted using a FACS Canto II instrument (BD Biosciences) or FACS Aria II instrument (BD Biosciences), respectively. Analysis of the flow cytometry data was done by FlowJo software (FlowJo v10). Gating strategies are provided in Supplementary Fig. 7a, b.

## Intracellular cytokine staining

To investigate cytokine production of adipose iNKT cells, WAT was digested as described above, and SVF was activated as described previously[32], with minor modifications. Briefly, SVF was cultured for 4–6 hours in the presence of phorbol 12-myristate 13-acetate (50 ng/ml, P8139, Thermo Fisher), Ionomycin (1 μg/ml, I0634, Thermo Fisher), and Brefeldin A (5 μg/ml, 420601, BioLegend). Cultures were in complete RPMI media [10% fetal bovine serum (FBS) (Young In Frontier), 1% penicillin/streptomycin (Welgene), 10 mM HEPES, 1% non-essential amino acid (Welgene), 1 mM sodium pyruvate, 50 μM β-mercaptoethanol, and 2 mM L-glutamine]. The cells were washed with PBS, stained and analyzed by FACS Canto II instrument as described above.

## Generation of primary iNKT cells

Primary iNKT cells were prepared as reported previously[90]. Briefly, the sorted TCRβ$^{int}$/CD1d.PBS57 tetramer$^+$ iNKT cells from spleen were stimulated with anti-CD3e (3 μg/mL, 14-0031-82, Thermo Fisher) and anti-CD28 antibodies (1 μg/mL, 10312-20, Biogems) for three days and then expanded with mouse recombinant IL-2 (10 ng/mL, 212-12, Peprotech) and IL-7 (10 ng/mL, 217-17, Peprotech) for 10 days in complete RPMI media described above. The culture media of primary iNKT cells were changed every two day.

## iNKT cell adoptive transfer

Primary iNKT cells (~5 × 10$^5$ cells/mouse), prepared as indicated above, were intravenously injected into 3- or 8-week-old male mice and analyzed after 8 weeks. In vivo-expanded Adipose iNKT cell subpopulations (donor cells) were purified by FACS from WAT of

WT or CD45.1 male mice 1 week after α-GC injection. Each adipose iNKT cell subpopulation was concentrated to ~1000 cells/µl by centrifugation. Donor cells (20 µl) were injected into the fat pads of WT C57BL/6 adult male mice. Donor CD45.1⁺ cells were harvested from the recipient animals 3 weeks after transplantation and subjected to FACS analysis.

### Coculture

3T3-L1 pre-adipocytes were grown to confluence and then differentiated in 96- or 48-well culture plates. During differentiation, the cells were incubated in adipogenic medium [Dulbecco's modified Eagle's medium (DMEM) containing 10% FBS, 1 µM dexamethasone, 520 µM Isobutylmethylxanthine, and 850 nM insulin]. Two days after adipogenic induction, the cells were treated with FI medium [DMEM containing 10% FBS and 850 nM insulin] for two days. Differentiated 3T3-L1 adipocytes were treated with FFA (500 µM) as previously described[41]. Briefly, FFAs (palmitic acid or oleic acid) were dissolved in ethanol and diluted in DMEM low glucose media containing 1% FBS and 2% BSA at 55 °C for 10 min. BSA-conjugated FFA-containing media were used for challenging differentiated 3T3-L1 adipocytes for 2 weeks and media were changed every two days. For 3T3-L1 adipocyte and iNKT cell coculture experiment, iNKT cell subpopulations were sorted after in vivo iNKT cell expansion as indicated above and stained with CellTracker™ Green CMFDA Dye (C2925, Thermo Fisher) at 37°C for 30 min. 10⁴ of stained iNKT cells were cocultured with FFA-challenged 3T3-L1 adipocytes with or without anti-CD1d antibody (20 µg/ml, BE0000, Bio X Cell) in 48 well plate. After 3 days of coculture, the cells were incubated with Hoechst 33342 (1:1000, H3570, Thermo Fisher) and propidium iodide (PI) (1:200, 51-66211E, Thermo Fisher) at 37°C for 20 min in the dark. The cells were imaged and quantified using a CQ1 microscope (Yokogawa). Diameters of lipid droplets were calculated by Image J software.

### Immunohistochemistry

Whole-mount imaging was carried out as previously described with minor modifications[18]. Whole epididymal WAT was fixed in 4% paraformaldehyde for overnight and blocked with PBS containing 5% horse serum and 0.3% Triton X-100 for 1 hour. The fixed tissues were incubated with primary antibodies against GFP (1:1000, NB100-62622, Novus Biologicals) and perilipin (1:1000, 20R-PP004, Fitzgerald) at 4°C overnight. The tissues were then washed three times for 10 min each, incubated with secondary antibodies (1:1000, A-21099, Thermo Fisher and 1:1000, ab6904, Abcam) and Hoechst 33342 at RT for 4 hours, washed three times for 10 min each, and mounted on slides in 8-well plates (155409, Nunc Lab-Tek II). Tissues were observed and imaged using a CQ1 microscope. Diameters of lipid droplets were calculated by Image J software.

For immunohistochemistry, small fractions of fat tissues were isolated from mice, fixed in 4% paraformaldehyde, and embedded in paraffin. The paraffin blocks were cut into 5-µm sections and stained with hematoxylin and eosin. Tissues were imaged using a digital slide scanner (Axio Scan Z1, Carl Zeiss).

### THP-1 monocyte migration assay

Synthetic CCL5 peptides (500 ng/ml, 250-07, Peprotech) were dissolved in serum-free RPMI1640 media and placed in 24-well culture plate. THP-1 cells were pre-stained with CellTracker™ Deep Red Dye (C34565, Thermo Fisher) for 30 min and 5 × 10⁴ (per well) THP-1 cells were loaded on the surface of the upper layer of Trans-well insert (8 µm pore, 3422, Corning). 6 hours after incubation, the upper layer and Trans-well insert were carefully removed. Migrated THP-1 cells were imaged and quantified using the CQ1 confocal microscope and cell counter (NanoEntek), respectively.

### Primary cell conditioned media (CM) preparation

Primary adipocytes were obtained from the floating fraction of fractionated epididymal WAT as described above and primary hepatocytes were isolated by collagenase perfusion method as described previously[91] from 12-to-16-week-old male mice. Primary cells were cultured in complete RPMI media for 2 days. CM was collected and frozen at −80°C until use.

### DN32.D3 cell culture

DN32.D3 iNKT hybridoma cell line was cultured in complete RPMI media. Adipocyte conditioned media (AD CM) and primary hepatocyte conditioned media (Hep CM), made as described above, were mixed in a 1:1 ratio with fresh complete RPMI media and treated to 2 × 10⁴ DN32.D3 cells in a 12-well plate. DN32.D3 cells were incubated for 2 days in CM and harvested.

### RNA isolation and RT-qPCR

For primary adipocytes, 3T3-L1 adipocytes, or DN32.D3 iNKT hybridoma, total RNA was isolated as previously described[18]. For sorted iNKT cells and in vivo-expanded adipose iNKT cell subpopulations, total RNA was isolated by using Direct-zol™ RNA MiniPrep (R2062, Zymo Research) following the manufacturer's protocol. The isolated RNA was reverse-transcribed using the ReverTra Ace qPCR RT Kit (Toyobo). RT-qPCRs were run using SYBR Green master mix (DQ384-40h, BioFact). Target gene expression levels were normalized to *RplpO* (36B4) expression. Primer sequences are listed in Supplementary Table 2.

### Statistical analysis

Data are presented as the mean ± standard deviation (SD). n-Values indicated in the figures refer to biological replicates. The means of the two groups were compared using a two-tailed Student's t-test. Means of multiple groups were compared using one-way ANOVA followed by Tukey's post-hoc test. Two independent variables were compared using two-way ANOVA followed by Sidak's multiple comparisons test. Statistical analyses were performed using GraphPad Prism (GraphPad Software v9).

## Data availability

The raw sequencing data from scRNA-seq of adipose and thymic iNKT cells have been deposited in NCBI accession code PRJNA917041. The processed data from scRNA-seq of adipose and thymic iNKT cells have been deposited in Zenodo database accession code 10121624. Further information and requests for resources and reagents should be directed to and will be fulfilled by the lead contact Jae Bum Kim (jaebkim@snu.ac.kr). Source data are provided with this paper.

## Code availability

Codes to reproduce results in this paper are uploaded [https://github.com/CB-postech/NATURE-COMMUNICATIONS-adipose-iNKT].

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

## Acknowledgements

We would like to thank Dr. Doo Hyun Chung for providing Jα18 KO and CD1d KO mice and Dr. Rho Hyun Seong for providing CD45.1 and 4Get mice. The authors thank the National Institutes of Health Tetramer Core Facility for the generous gift of CD1d tetramers. We are grateful to Hye Jin Noh for helping with the FACS experiments. This study was supported by the National Research Foundation, funded by the Korean government (Ministry of Science and ICT; NRF-2021M3H9A1030158, NRF-2023R1A2C3004065, and NRF- 2021R1A6A1A10042944 to J.K.K. and NRF-2018R1A5A1024340, NRF-2020R1A3B2078617, and RS-2023-00218616 to J.B.K.). S.M.H., Je.P., H.N., K.M.Y., W.T.L., Y.G.J., and K.C.S. were supported by the BK21 Plus program.

## Author contributions

S.M.H. and Je.P. conceptualized the study, performed the experiments, analyzed the data, and wrote the manuscript. E.S.P. and Y.H.C. analyzed

the scRNA-seq data and wrote the manuscript. H.N., J.O., Ji.P., K.M.Y., and W.T.L. performed the animal experiments. Y.K.L. and S.H.C. performed the experiments on human adipose tissue. Y.G.J., K.C.S., and J.Y.H. discussed the study and contributed to the writing of the manuscript. J.K.K. supervised the bioinformatics analyses and wrote the manuscript. J.B.K. supervised the study and wrote the manuscript.

## Competing interests

The authors declare no competing interests.
