## [Peer Review File · Nature Communications]

REVIEWER COMMENTS

Reviewer #1 (expert in iNKT cells):

In this manuscript Han et al have investigated the transcriptional profile of murine adipose iNKT cells and in the context of obesity, characterized the potential roles of iNKT subsets in the murine models. This study is of interest to understand the immune regulation that occurs in the fat by unconventional T cells and has relevance to harness the transcriptional profile of adipose-specific T cells iNKT cells in disease. I have general and specific comments regarding analyses of the data and clarification/justification of the analyses employed.

Regarding the methods for scRNAseq, there seems to be a lack of clarification for what samples were used for scRNAseq analyses and how/what tissues were prepared, sorted etc. It's unclear why tissue samples from liver, spleen and LNs were not included in the initial Seurat UMAP in Fig1, would it not be useful to see how these tissues integrate with the adipose and thymus?

Understanding what is differentially expressed between NKT17 in thymus vs adipose and TH17 and also NKT1 in thymus vs NKT1 in adipose seems important as these subsets are define as the same subset, yet do not cluster together. Such differentially expressed genes may also contribute to a gene expression program signature that the authors could calculate.

Regarding the adoptive transfer experiments in the main text, it's not clear what tissue the source of the primary iNKT cells is given that subsets differ/tissue this may affect how the microenvironment impacts the KLRG upregulation and other functional properties. It's not clear what in the adipocyte condition media is causing KLRG to go up? Have the authors analysed this by protein/bead array? Has it been shown that iNKT cells can recognize lipid overloaded adipocytes in a TCR/CD1d dependent mechanism?

Regarding the Areg experiments, it's not quite directly showing that NKT-derived Areg will elicit those effects, rather what Areg can do. Is there dose dependency for those experiments? What about Areg derived from other immune subsets in the fat, can it be expressed by others? This doesn't seem to be discussed, how do Areg levels from NKT cells compare to other immune cells?

Reviewer #2 (expert in adipose tissue transcriptomics and genomics):

In this paper, Han, Park, and Park et al describe subpopulations of iNKT cells in the adipose tissue and specifically identify an adipose tissue-selective iNKT cell population. They additionally show that a population of cytotoxic iNKT cells arises from these adipose selective iNKT cells during HFD conditions and mediate adipocyte death. Finally, they show that a different population of iNKT cells mediate ASC proliferation through secretion of Areg. Overall, this paper convincingly identifies a number of distinct iNKT populations in the adipose tissue and dives into some of the ways these subpopulations mediate adipose tissue in the disease state. The findings in this paper are novel and noteworthy. However, there are some lingering mechanistic questions that, if answered, could enhance the paper, as well as a lack of detailed experimental information in some areas, particularly about the single cell sequencing experiments. Specific comments are as follows:

1. The authors convincingly show that something about the adipose tissue microenvironment stimulates iNKT cells to become A*s*-iNKT1 cells, and specifically the experiments shown in Figure 2 J&K suggest that it may be a secreted factor. The experiments on Cd1d1 KO mice suggest that this is not the mediator, what are other possible candidates for this mediator? Perhaps the authors could try an in vitro screen of some potential candidates.

2. The data supporting the model of A-iNKT17 secreted AREG leading to ASC proliferation is suggestive but could be more direct. For example, repeating the experiment in 5e but including A-iNKT17 cells in which AREG has been knocked down. Without direct evidence that the AREG is coming from the iNKT17 cells (what about the non-iNKT cells in the adipose tissue?) the experiments in 5i-m fall a little flat.

3. In the flow cytometry experiments in Figure 5, ASCs are defined by a negative- ie CD45- and CD31-. Please also use a positive marker of ASCs such as PDGFRa.

4. There is no scRNA-seq sample preparation section in the methods. Please provide this, and include information on which adipose depot(s) were sequenced and if any cell sorting occurred before sequencing (and if so, what cell sorting). If the later sections such as 'adipose tissue fractionation' are meant to serve as scRNA-seq sample preparation sections, please make this clearer.

5. The projection analysis in Figure 1h-k is difficult to interpret, as many cells seem to map outside the bounds of the UMAP, and because there is no information on what clusters the cells map to, just the projected UMAP coordinate. The authors should try alternate methods of combining these datasets to better show what clusters each cell maps to, such as integration and/or reference mapping (such as is available in Seurat v4).

6. Similarly, from the methods it does not appear that there was any integration either across adipose samples or between adipose and thymus (as seen in figure 1a). If the samples are integrated, is there more harmony across tissue?

7. Ja18 KO mice are used for multiple studies in this paper but these mice are not introduced. Please briefly explain what this model is.

8. Please be sure to make **processed** scRNA-seq data available online (at the time of publication), ideally on a platform designed to easily query this kind of dataset such as cellxgene (<https://cellxgene.cziscience.com/>) and/or the single cell portal (https://singlecell.broadinstitute.org/single_cell).

9. Please clarify which WAT depot(s) are used in experiments throughout the paper.

Reviewer #3 (expert in scRNAseq of adipose tissue):

In this research, author found the distinct subpopulations of adipose iNKT cells which modulating adipocyte death and birth for adipose tissue homeostasis and each adipose iNKT subpopulation plays key roles to turnover of adipocyte by crosstalk in adipose tissue. The results of the article are very interesting.

1. The ssc-a values in Supplementary Figure 6a-Lymphocytes can be appropriately changed.

2. Please supplement the depth scales of color in Figure 4a and Extended Data Figure 2k (results of KEGG).

3. In Figure 4d (coculture experiment), there are two identical "merge", please adjust them.

RESPONSE TO REVIEWERS' COMMENTS

Reviewer #1 (expert in iNKT cells):

In this manuscript Han et al have investigated the transcriptional profile of murine adipose iNKT cells and in the context of obesity, characterized the potential roles of iNKT subsets in the murine models. This study is of interest to understand the immune regulation that occurs in the fat by unconventional T cells and has relevance to harness the transcriptional profile of adipose-specific T cells iNKT cells in disease. I have general and specific comments regarding analyses of the data and clarification/justification of the analyses employed.

Regarding the methods for scRNAseq, there seems to be a lack of clarification for what samples were used for scRNAseq analyses and how/what tissues were prepared, sorted etc. It's unclear why tissue samples from liver, spleen and LNs were not included in the initial Seurat UMAP in Fig1, would it not be useful to see how these tissues integrate with the adipose and thymus? Understanding what is differentially expressed between NKT17 in thymus vs adipose and TH17 and also NKT1 in thymus vs NKT1 in adipose seems important as these subsets are define as the same subset, yet do not cluster together. Such differentially expressed genes may also contribute to a gene expression program signature that the authors could calculate.

Regarding the adoptive transfer experiments in the main text, it's not clear what tissue the source of the primary iNKT cells is given that subsets differ/tissue this may affect how the microenvironment impacts the KLRG upregulation and other functional properties. It's not clear what in the adipocyte condition media is causing KLRG to go up? Have the authors analysed this by protein/bead array? Has it been shown that iNKT cells can recognize lipid overloaded adipocytes in a TCR/CD1d dependent mechanism?

Regarding the Areg experiments, it's not quite directly showing that NKT-derived Areg will elicit those effects, rather what Areg can do. Is there dose dependency for those experiments? What about Areg derived from other immune subsets in the fat, can it be expressed by others? This doesn't seem to be discussed, how do Areg levels from NKT cells compare to other immune cells?

Reviewer #2 (expert in adipose tissue transcriptomics and genomics):

In this paper, Han, Park, and Park et al describe subpopulations of iNKT cells in the adipose tissue and specifically identify an adipose tissue-selective iNKT cell population. They additionally show that a population of cytotoxic iNKT cells arises from these adipose selective iNKT cells during HFD conditions and mediate adipocyte death. Finally, they show that a different population of iNKT cells mediate ASC proliferation through secretion of Areg. Overall, this paper convincingly identifies a number of distinct iNKT populations in the adipose tissue and dives into some of the ways these subpopulations mediate adipose tissue in the disease state. The findings in this paper are novel and noteworthy. However, there are some lingering mechanistic questions that, if answered, could enhance the paper, as well as a lack of detailed experimental information in some areas, particularly about the single cell sequencing experiments. Specific comments are as follows:

1. The authors convincingly show that something about the adipose tissue microenvironment stimulates iNKT cells to become As-iNKT1 cells, and specifically the experiments shown in Figure 2 J&K suggest that it may be a secreted

factor. The experiments on Cd1d1 KO mice suggest that this is not the mediator, what are other possible candidates for this mediator? Perhaps the authors could try an in vitro screen of some potential candidates.

2. The data supporting the model of A-iNKT17 secreted AREG leading to ASC proliferation is suggestive but could be more direct. For example, repeating the experiment in 5e but including A-iNKT17 cells in which AREG has been knocked down. Without direct evidence that the AREG is coming from the iNKT17 cells (what about the non-iNKT cells in the adipose tissue?) the experiments in 5i-m fall a little flat.

3. In the flow cytometry experiments in Figure 5, ASCs are defined by a negative- ie CD45- and CD31-. Please also use a positive marker of ASCs such as PDGFRa.

4. There is no scRNA-seq sample preparation section in the methods. Please provide this, and include information on which adipose depot(s) were sequenced and if any cell sorting occurred before sequencing (and if so, what cell sorting). If the later sections such as 'adipose tissue fractionation' are meant to serve as scRNA-seq sample preparation sections, please make this clearer.

5. The projection analysis in Figure 1h-k is difficult to interpret, as many cells seem to map outside the bounds of the UMAP, and because there is no information on what clusters the cells map to, just the projected UMAP coordinate. The authors should try alternate methods of combining these datasets to better show what clusters each cell maps to, such as integration and/or reference mapping (such as is available in Seurat v4).

6. Similarly, from the methods it does not appear that there was any integration either across adipose samples or between adipose and thymus (as seen in figure 1a). If the samples are integrated, is there more harmony across tissue?

7. Ja18 KO mice are used for multiple studies in this paper but these mice are not introduced. Please briefly explain what this model is.

8. Please be sure to make **processed** scRNA-seq data available online (at the time of publication), ideally on a platform designed to easily query this kind of dataset such as cellxgene (<https://cellxgene.cziscience.com/>) and/or the single cell portal (https://singlecell.broadinstitute.org/single_cell).

9. Please clarify which WAT depot(s) are used in experiments throughout the paper.

Reviewer #3 (expert in scRNAseq of adipose tissue):

In this research, author found the distinct subpopulations of adipose iNKT cells which modulating adipocyte death and birth for adipose tissue homeostasis and each adipose iNKT subpopulation plays key roles to turnover of adipocyte by crosstalk in adipose tissue. The results of the article are very interesting.

1. The ssc-a values in Supplementary Figure 6a-Lymphocytes can be appropriately changed.
2. Please supplement the depth scales of color in Figure 4a and Extended Data Figure 2k (results of KEGG).
3. In Figure 4d (coculture experiment), there are two identical "merge", please adjust them.

Response to the reviewers' comments
manuscript NCOMMS-23-13999

Unique Adipose Invariant Natural Killer T Cell Subpopulations Control Adipocyte Turnover by Orchestrating Cellular Crosstalk

Reviewer #1 (expert in iNKT cells):

In this manuscript Han et al have investigated the transcriptional profile of murine adipose iNKT cells and in the context of obesity, characterized the potential roles of iNKT subsets in the murine models. This study is of interest to understand the immune regulation that occurs in the fat by unconventional T cells and has relevance to harness the transcriptional profile of adipose-specific T cells iNKT cells in disease. I have general and specific comments regarding analyses of the data and clarification/justification of the analyses employed.

Regarding the methods for scRNAseq, there seems to be a lack of clarification for what samples were used for scRNAseq analyses and how/what tissues were prepared, sorted etc. It's unclear why tissue samples from liver, spleen, and LNs were not included in the initial Seurat UMAP in Fig1, would it not be useful to see how these tissues integrate with the adipose and thymus? Understanding what is differentially expressed between NKT17 in thymus vs adipose and TH17 and also NKT1 in thymus vs NKT1 in adipose seems important as these subsets are defined as the same subset, yet do not cluster together. Such differentially expressed genes may also contribute to a gene expression program signature that the authors could calculate.

We appreciate these comments. To clarify the source of iNKT cells and the tissue preparation process for scRNA-seq, we depicted these processes in more detail in new Extended Data Fig. 1a (Lines 851–857), Methods (Lines 484–487), and Results (Lines 76–77) sections. Briefly, we sorted TCR β^{int} and CD1d.PBS57 tetramer⁺ iNKT cells from epididymal white adipose tissue (WAT) of NCD-, 1w HFD-, 8w HFD-fed 16-week-old male C57BL/6 mice and thymus of NCD-fed mice by FACS. These cells were then subjected to scRNA-seq analysis.

As suggested by the reviewer, to examine how iNKT cells from different organs integrate with each other, we performed clustering analysis with the combined dataset consisting of our in-house and public scRNA-seq data. As shown in below Reviewer's only Figure 1a, cells were clustered according to each tissue without batch correction. Thus, we corrected for batch effects by considering tissue as a batch (Reviewer's only Figure 1b). Adipose and thymic iNKT cells from our in-house dataset were separated from iNKT cells from other organs, which could be due to differences in experimental platforms between the two datasets (5' gene expression libraries (adipose and thymus) and 3' gene expression libraries (liver, spleen, and lymph nodes)). Since batch effects due to platform differences were not corrected, we used adipose tissue iNKT cells as a reference and compared iNKT cells from other organs using projection and reference mapping analysis (Fig. 1h–l).

Reviewer's only Figure 1

a, Unsupervised clustering of iNKT cells from various organs/tissues on a uniform manifold approximation and projection (UMAP) plot. iNKT scRNA-seq data was obtained from our study (adipose and thymic iNKT cells) and GSE161495 (splenic, hepatic, and lymph node iNKT cells). **b**, Batch correction was conducted in (a).

To understand which genes are differentially expressed in adipose and thymic iNKT cells, we analyzed differentially expressed genes (DEGs) between adipose iNKT1 cells (A-iNKT1) and thymic iNKT1 cells (T-iNKT1). Also, we compared adipose iNKT17 cells (A-iNKT17) and thymic iNKT17 cells (T-iNKT17) (new Extended Data Fig. 1b-g). A-iNKT1 cells expressed relatively high levels of previously reported adipose iNKT cell transcription factor (*Nfil3*)^{1,2}, antigen presentation-related genes (*H2-Q6* and *H2-Q7*), T cell receptor signaling pathway-related genes (*Lcp2*, *Itk*, *Rasgrp1*, *Ifng*, and *Tnf*), and cytoskeleton and focal adhesion-related genes (*Actb*, *Actg1*, *Itga4*, *Itgb1*, *Itgb7*, and *Rac2*), while A-iNKT1 cells expressed relatively low levels of ribosomal genes (*Rpl21* and *Rpl39*), oxidative phosphorylation (OXPHOS)-related genes (*mt-Co1*, *mt-Atp8*, *mt-Cytb*, *Ndufa3*, *Ndufa4*, and *Atp6v0e*), and several NK and cytokine receptor genes (*Klra9*, *Klrb1c*, and *Il7r*) compared to T-iNKT1 cells (new Extended Data Fig. 1b-d).

A-iNKT17 cells showed abundant gene expression levels of antigen presentation-related genes (*H2-D1*, *H2-K1*, and *H2-Q7*), cellular senescence, and pro-survival genes (*Cdkn1a*, *Cnd3*, *Gadd45b*, and *Bcl2*), while exhibiting lower gene expression levels of ribosomal genes (*Rpl28* and *Rps26*), Th17 cell differentiation-related genes (*Il1r1*, *Il23r*, *Rorc*, *Zap70*, and *Il22*), and OXPHOS-related genes (*mt-Atp8* and *mt-Cytb*) compared to T-iNKT17 cells (new Extended Data Fig. 1e-g).

To summarize, adipose iNKT cells showed higher gene expression profiles of antigen presentation and apoptosis/senescence-related genes while thymic iNKT cells exhibited higher gene expression profiles of OXPHOS and ribosome-related genes. The meaning of differentially regulated pathways between adipose and thymic iNKT cells and their functional differences require further investigation. This DEG analysis was attached to new Extended Data Fig. 1 and described in the revised manuscript (Lines 83–84, 857–861). In addition, DEGs are listed in new Supplementary Table 3.

Regarding the adoptive transfer experiments in the main text, it's not clear what tissue the source of the primary iNKT cells is given that subsets differ/tissue this may affect how the microenvironment impacts the KLRG upregulation and other functional properties. It's not clear what in the adipocyte condition media is causing KLRG to go up? Have the authors analysed this by protein/bead array? Has it been shown that iNKT cells can recognize lipid overloaded adipocytes in a TCR/CD1d dependent mechanism?

Thank you for these critiques and suggestions. We used TCR β^{int} /CD1d.PBS57 tetramer⁺ splenic iNKT cells to make primary iNKT cells for *in vivo* studies. We also added an explanation about the tissue source of primary iNKT cells in Results section (Line 140), Figure Legend (Line 744), and Methods section (Line 623).

Identifying the adipocyte-secreting factor(s) to induce As-iNKT1 cell generation is an interesting and important topic and worth to be further investigated. To address the reviewer's question, we conducted several experiments as described below (Reviewer's only Figure 2). To test whether the factor(s) to induce As-iNKT1 cells might be protein, WAT primary adipocyte-conditioned media (AD CM) were heat-inactivated at 95°C for 10 minutes (AD CM HI) and then treated to iNKT cells. Interestingly, heat-inactivation showed negligible effects on As-iNKT1-high gene upregulation or Au-iNKT1-high gene downregulation (Reviewer's only Figure 2a,b). This result implies that As-iNKT1-stimulating factor(s) from adipocytes might be heat-inactivation-insensitive factors such as lipids. To investigate whether lipids in AD CM might be crucial for the induction of As-iNKT1 cell characteristics, AD CM were delipidated with butanol/di-isopropyl ether method (AD CM delip)³. As shown in Reviewer's only Figure 2c, this method decreased the level of lipid in AD CM. Delipidation somewhat decreased the effects of AD CM on the induction of As-iNKT1 cell characteristics (Reviewer's only Figure 2d). To further characterize which lipid species from adipocytes would promote As-iNKT1 cell generation, iNKT cell lines, DN32.D3, were treated with various lipids including free fatty acids and lipokines⁴ (Reviewer's only Figure 2e,f). Among these, OA (oleate) appeared to induce KLRG1 expression, a selective surface marker for As-iNKT1 cells, and As-iNKT1 cell-related transcription patterns.

Other lipid species and underlying mechanisms related to As-iNKT1 generation should be investigated in the future. On the other hand, when well-known adipokines, adiponectin and leptin, as a part of adipocyte-releasing proteins, were tested, they had no effects on As-iNKT1 cell generation (Reviewer's only Figure 2e,f). Currently, we have been working on this issue as an independent follow-up project (as mentioned in lines 365–366). Thus, we would like to share preliminary data below only for the reviewers.

Reviewer's only Figure 2

a, mRNA levels of *Bhlhe40*, *Rgs1*, *Maf*, *Nr4a1*, *Cd226*, *Satb1*, *Il7r*, and *Klif2* in DN32.D3 iNKT hybridoma cells after 2 days of culture with Control media (Ctrl), WAT primary adipocyte-conditioned media (AD CM), AD CM with heat inactivation (AD CM HI), or primary hepatocyte-conditioned media (Hep CM) (n = 5–6). **b**, Volcano plot of DEGs between As-iNKT1 and Au-iNKT1 cells. **c**, Concentration of triglyceride (TG) in Ctrl, AD CM, and delipidated adipocyte-conditioned media (AD CM delip) (n = 3, technical replicates). **d**, mRNA levels of *Rgs1*, *Maf*, *Cd226*, *Satb1*, *Il7r*, and *Klif2* in DN32.D3 iNKT hybridoma cells after 2 days of culture with Ctrl, AD CM, or AD CM delip (n = 6). **e**, Representative FACS plots and the ratio of KLRG1⁺ cells in DN32.D3 cells treated with various lipids or adipokines for 2 days (n = 3). Free fatty acids and lipokines were conjugated with 2% BSA at 55°C for 10 minutes. **f**, mRNA levels of *Rgs1*, *Maf*, *Cd226*, *Satb1*, *Il7r*, and *Klif2* in DN32.D3 iNKT hybridoma cell lines after 2 days of culture with various lipids or adipokines (n = 5–6). Data are represented as mean ± SD. n.s., non-significant. One-way ANOVA (**a**, and **c–e**).

PA; palmitate, 200 μ M, **PO**; palmitoleate, 200 μ M, **SA**; stearate, 200 μ M, **OA**; oleate, 200 μ M, **C18:1 LPA**; lysophosphatidic acid, 20 μ M, **5-PAHSA**; palmitic acid esters of hydroxystearic acid, 20 μ M, **Adiponectin**; 5 μ g/ml, **Leptin**; 250 ng/ml.

To answer the reviewer's question whether iNKT cells recognize lipid overloaded adipocytes via TCR/CD1d, we cocultured Ac-iNKT1 cells and PA-treated 3T3-L1 adipocytes with or without CD1d neutralizing antibody (new Extended Data Fig. 5n). The ratio of dead adipocytes was slightly but significantly downregulated in the presence of CD1d neutralizing antibody, implying that TCR-CD1d interaction would contribute to the killing of enlarged and inflamed adipocytes by Ac-iNKT1 cell. This result and experimental condition were described in lines 264–267, 604–605, and 938–940.

Regarding the Areg experiments, it's not quite directly showing that NKT-derived Areg will elicit those effects, rather what Areg can do. Is there dose dependency for those experiments? What about Areg derived from other immune subsets in the fat, can it be expressed by others? This doesn't seem to be discussed, how do Areg levels from NKT cells compare to other immune cells?

We appreciate these comments. Following the reviewer's suggestion, we tested the dose response of AREG on ASC proliferation *in vivo*. We found that AREG upregulated ASC proliferation in a dose-dependent manner (new Fig. 5h, Lines 315–316, 826–828). As the reviewers pointed out, we also agree that the source of AREG in WAT is an important question that needs to be discussed given that AREG can be produced by different cell types such as Treg cells, ILC2, Th2 cells, and macrophages⁵⁻⁸. In WAT, ILC2, Treg cells, and Th2 CD4⁺ T cells appear to express AREG according to previous reports and public scRNA-seq data^{7,9}. To compare AREG production in various immune cell types, intracellular cytokine staining has been conducted in A-iNKT17 cells, Treg cells, and ILC2 from 12-week HFD-fed male mice. A-iNKT17 cells expressed AREG and had a similar proportion of AREG-positive cells compared to Treg cells and ILC2 (Reviewer's only Figure 3a,b). This result suggests that A-iNKT17 cells would not be the only cell type expressing AREG in adipose tissue and A-iNKT17 cells would promote ASC proliferation potentially by cooperating with other immune cell types expressing AREG. We discussed this in Discussion section (Lines 404–410).

Reviewer's only Figure 3

a, The ratio of AREG-positive cells among indicated cell types and representative FACS plot of intracellular cytokine staining. Adipose tissue stromal vascular fraction (SVF) from 12-week HFD-fed male mice was incubated in complete RPMI media containing Brefeldin A (10 μg/ml) with or without stimulating reagents (100 ng/ml of PMA and 1 μg/ml of Ionomycin) for 5 hours. Trapped AREG proteins inside cells were stained and analyzed by FACS in A-iNKT17 cells, Treg cells, and ILC2 (n = 4). **b**, Gating strategy for A-iNKT17 cells, Treg cells, and ILC2 in adipose tissue SVF. Data are represented as mean ± SD.

Reviewer #2 (expert in adipose tissue transcriptomics and genomics):

In this paper, Han, Park, and Park et al describe subpopulations of iNKT cells in the adipose tissue and specifically identify an adipose tissue-selective iNKT cell population. They additionally show that a population of cytotoxic iNKT cells arises from these adipose selective iNKT cells during HFD conditions and mediate adipocyte death. Finally, they show that a different population of iNKT cells mediate ASC proliferation through secretion of Areg. Overall, this paper convincingly identifies a number of distinct iNKT populations in the adipose tissue and dives into some of the ways these subpopulations mediate adipose tissue in the disease state. The findings in this paper are novel and noteworthy. However, there are some lingering mechanistic questions that, if answered, could enhance the paper, as well as a lack of detailed experimental information in some areas, particularly about the single cell sequencing experiments. Specific comments are as follows:

1. The authors convincingly show that something about the adipose tissue microenvironment stimulates iNKT cells to become As-iNKT1 cells, and specifically the experiments shown in Figure 2 J&K suggest that it may be a secreted factor. The experiments on Cd1d1 KO mice suggest that this is not the mediator, what are other possible candidates for this mediator? Perhaps the authors could try an in vitro screen of some potential candidates.

We appreciate these questions. In this study, we would like to suggest that adipocyte-derived secretory factors could promote As-iNKT1 generation with the data from treating adipocyte-conditioned media (AD CM) to DN32.D3 iNKT hybridoma cell line. To address the reviewer's question, we tried several ways to characterize the nature of As-iNKT1 cell stimulating factors in AD CM. At the beginning, we did heat inactivation of AD CM at 95°C for 10 minutes to test whether proteins might be involved in As-iNKT1 cell generation. As shown in Reviewer's only Figure 2a, heat inactivation of AD CM showed negligible effects on gene expressions of DN32.D3 cells. This potentially proposes that heat inactivation-insensitive factors including lipids might be involved in stimulating As-iNKT1 generation. Then, to investigate whether lipids might affect As-iNKT1 cell generation, AD CM was delipidated with butanol and di-isopropyl ether (AD CM delip)³. As shown in Reviewer's only Figure 2c, this method decreased the level of lipid in AD CM. AD CM delip had somewhat reduced effects on the gene expression of DN32.D3 cells compared to AD CM (Reviewer's only Figure 2d). With these data, we are able to assume that lipids secreted from adipocytes would promote As-iNKT1 generation. Due to extremely diverse lipid species and short revision periods, we only tested several lipokines⁴ and free fatty acids, and investigated whether these lipids could upregulate or downregulate As-iNKT1 or Au-iNKT1 marker genes, respectively. As shown in Reviewer's only Figure 2e and 2f, OA (oleate) appeared to induce As-iNKT1 cell characteristic gene expressions such as KLRG1, *Maf*, *Rgs1*, and *Cd226*. Other lipid species and underlying mechanisms that govern As-iNKT1 cell generation should be further investigated as an independent project. We mentioned that this issue will be investigated in the near future (Lines 365–366). In addition, well-known adipokines (adiponectin and leptin) did not show any effects on As-iNKT1 cell generation (Reviewer's only Figure 2e,f).

2. The data supporting the model of A-iNKT17 secreted AREG leading to ASC proliferation is suggestive but could be more direct. For example, repeating the experiment in 5e but including A-iNKT17 cells in which AREG has been knocked down. Without direct evidence that the AREG is coming from the iNKT17 cells (what about the non-iNKT cells in the adipose tissue?) the experiments in 5i-m fall a little flat.

We really appreciate this critique. Previous reports and public scRNA-seq dataset suggest that AREG could be secreted by several cell types in WAT including ILC2, Treg cells, and CD4⁺ Th2 cells^{7,9}. Intracellular cytokine staining of AREG also showed that not only A-iNKT17 cells but also ILC2 or Treg cells could express AREG in epididymal WAT (Reviewer's only Figure 3a). To examine whether AREG from A-iNKT17 cells could stimulate ASC proliferation, we conducted several experiments. At first, as the reviewer suggested, we attempted to inject A-iNKT17 cells with reduced AREG into obese mice. To carry out this, we sorted out

in vivo-expanded A-iNKT17 cells from epididymal WAT and suppressed *Areg* via siRNA. Unfortunately, sorted A-iNKT17 cells rapidly died *ex vivo*. Instead, we decided to adopt alternative approaches. As a secreted signaling molecule, AREG should bind to EGFR receptors on its target cell¹⁰. To verify whether iNKT cell-induced ASC proliferation might be dependent on AREG-EGFR signaling, Gefitinib, a well-known EGFR inhibitor, was administered into WT male mice (as described in line 469) in the presence of α -GC that is a potent iNKT cell stimulator. As shown in new Extended Data Fig. 6e, an increase in ASC proliferation by iNKT cell activation was attenuated by EGFR inhibition. This result suggested that EGFR signaling would be crucial for ASC proliferation process induced by iNKT cell activation. Next, to further examine whether AREG secreted from A-iNKT17 could stimulate ASC proliferation, we injected A-iNKT17 cells into iNKT cell-deficient mice (*J α 18 KO*) and blocked EGFR signaling by administering Gefitinib (Reviewer's only Figure 4a). Interestingly, the effect of A-iNKT17 cells on ASC proliferation showed a blunted tendency in the Gefitinib-treated group (Reviewer's only Figure 4b). These data propose that A-iNKT17 would promote ASC proliferation via EGFR signaling although we could not test the direct effects of AREG secreted from A-iNKT17 cells due to the technical limitations. Thus, we toned down our suggestion that A-iNKT17 cells would regulate ASC proliferation (Line 413). New Extended Data Fig. 6e was described in lines 316–317 and 957–958, and the possibility that AREG could be produced in non-iNKT cells was also described in Discussion section (Lines 404–410).

Reviewer's only Figure 4

a, Experimental scheme of A-iNKT17 cell injection and subsequent EGFR inhibition via Gefitinib injection. **b**, The ratio of Ki-67⁺ cells among ASCs (CD31⁻/CD45⁻/PDGFR α ⁺). Each pair of connected dots represents paired tissues from a single mouse ($n = 3-4$). n.s., non-significant. Paired Two-way ANOVA (**b**).

3. In the flow cytometry experiments in Figure 5, ASCs are defined by a negative- ie CD45⁻ and CD31⁻. Please also use a positive marker of ASCs such as PDGFR α .

According to this suggestion, we analyzed ASCs using the PDGFR α antibody (new Extended Data Fig. 7b, staining condition in lines 604–605) and re-performed the experiments in Fig. 5 (new Figs. 5a,b,d,h, Lines 295, 298, 305, and 315–316). There were no significant differences between CD31⁻/CD45⁻ cells and CD31⁻/CD45⁻/PDGFR α ⁺ cells in the aspect of iNKT cell-induced ASC proliferation. Results which defined ASCs by CD31⁻/CD45⁻ in a previous manuscript were moved to Extended Data Fig. 6 (Fig. 5b,e \rightarrow Extended Data Fig. 6b,c, Lines 296 and 306). In the revised manuscript, CD31⁻/CD45⁻ cells and CD31⁻/CD45⁻/PDGFR α ⁺ cells were denoted as 'lineage (Lin)-negative cells' and 'ASCs', respectively (new Fig. 5a,b,d,h, and Extended Data Fig. 6b–d, 7b, Lines 817, 951, 953, 955, 957–958).

4. There is no scRNA-seq sample preparation section in the methods. Please provide this, and include information on which adipose depot(s) were sequenced and if any cell sorting occurred before sequencing (and if so, what cell sorting). If the later sections such as 'adipose tissue fractionation' are meant to serve as scRNA-seq sample preparation sections, please make this clearer.

Thank you for this comment. We isolated TCR β ^{int} and CD1d.PBS57 tetramer⁺ iNKT cells from epididymal white adipose tissue and thymus by flow cytometry. Then, these cells were subjected to scRNA-

seq analysis. We depicted the source of iNKT cells and tissue preparation process for scRNA-seq in new Extended Data Fig. 1a (Lines 851–857), Methods (Lines 484–487), and Results (Lines 76–77). Adipose tissue and thymus were processed as described in “Adipose tissue fractionation” and “Lymphocyte Preparation from Liver, Spleen, Thymus, and Blood” from the Methods section, respectively.

5. The projection analysis in Figure 1h-k is difficult to interpret, as many cells seem to map outside the bounds of the UMAP, and because there is no information on what clusters the cells map to, just the projected UMAP coordinate. The authors should try alternate methods of combining these datasets to better show what clusters each cell maps to, such as integration and/or reference mapping (such as is available in Seurat v4).

Thanks for this helpful suggestion. We performed a reference mapping analysis using Seurat v4. By using adipose iNKT cells as a reference, we mapped other iNKT cells to specific adipose iNKT cell subpopulations. For clarity, we also introduced a bar graph to illustrate the mapping results (new Fig. 1l, Lines 102 and 731–735). This approach was able to map most iNKT cells from different organs to specific adipose iNKT cell subpopulations, namely A1 (adipose universal iNKT1) and A3 (adipose iNKT17), consistent with the projection analysis in Fig. 1h–k.

6. Similarly, from the methods it does not appear that there was any integration either across adipose samples or between adipose and thymus (as seen in figure 1a). If the samples are integrated, is there more harmony across tissue?

We appreciate this comment. We did not perform scRNA-seq data integration to better capture tissue-specific cellular and molecular characteristics. To address the reviewer’s comment, we performed an integration analysis of adipose and thymic iNKT cells to correct for batch effects using Harmony (please see below Reviewer’s only Figure 5). Notably, both adipose and thymic iNKT17 cells were clustered together after harmony batch correction, implying that iNKT17 cells would show greater similarity across the tissues. On the other hand, Au-iNKT1 and T-iNKT2 were partially overlapped (Reviewer’s only Figure 5), while they have distinct molecular characteristics (Fig. 1b). To prevent any confusion, we would like to provide the integration result as a Reviewer’s only Figure.

Reviewer’s only Figure 5

Harmony batch correction of adipose and thymic iNKT cells from NCD-fed mice.

7. *Ja18 KO mice are used for multiple studies in this paper but these mice are not introduced. Please briefly explain what this model is.*

We apologize for the insufficient information of *Ja18* mice. Since almost every iNKT cell possesses an alpha chain composed of *V α 14-J α 18*, *Ja18* KO mouse has been considered as an iNKT cell-deficient model¹¹. We added a brief explanation of this in Results section (Lines 230, 269, and 296).

8. *Please be sure to make processed scRNA-seq data available online (at the time of publication), ideally on a platform designed to easily query this kind of dataset such as cellxgene (<https://cellxgene.cziscience.com/>) and/or the single cell portal (https://singlecell.broadinstitute.org/single_cell).*

We have uploaded the processed scRNA-seq data to Zenodo (<https://zenodo.org/record/8320751>).

9. *Please clarify which WAT depot(s) are used in experiments throughout the paper.*

In this study, we used epididymal WAT and described WAT depot in new Extended Data Fig. 1a, Results (Line 76), Methods (Lines 485, 553, 660, and 683), and Figure Legends section (Lines 719 and 853).

Reviewer #3 (expert in scRNAseq of adipose tissue):

In this research, author found the distinct subpopulations of adipose iNKT cells which modulating adipocyte death and birth for adipose tissue homeostasis and each adipose iNKT subpopulation plays key roles to turnover of adipocyte by crosstalk in adipose tissue. The results of the article are very interesting.

1. *The ssc-a values in Supplementary Figure 6a-Lymphocytes can be appropriately changed.*

We appreciate this comment. We have carefully examined our lymphocyte gating from previous Extended Data Fig. 6a (now Extended Data Fig. 7a in revised manuscript, Reviewer's only Figure 6a). Lymphocyte gating was determined based on cellular density near lymphocyte area and localization of $\alpha\beta$ T cells, one of the representative lymphocytes, in FSC-A/SSC-A plot. As shown in the Reviewers' only Figure 6b, our gating could detect the majority of live $\alpha\beta$ T cells. Also, our lymphocyte gating looks similar to previous findings that investigated immune cells in white adipose tissue^{2,12}. Finally, when the lymphocyte gating was heightened, iNKT cell gating easily became indistinct because of autofluorescence from the cells with higher granularity (Reviewer's only Figure 6c). Based on these, we carefully concluded that the previous lymphocyte gating was appropriate for detecting adipose tissue iNKT cells.

Reviewer's only Figure 6

a, Gating Strategy of adipose iNKT cells from epididymal WAT SVFs. **b**, Gating of $\alpha\beta$ T cells and their localization in FSC-A/SSC-A plot. **c**, Lymphocyte gating with heightened SSC-A and subsequent iNKT cell gating. Red dotted circle indicates autofluorescence signal from cells with higher granularity.

2. Please supplement the depth scales of color in Figure 4a and Extended Data Figure 2k (results of KEGG).

We apologize for the confusion. Colored bars were merely an emphasis on specific KEGG terms. We unified the color code in a single bar graph (Fig. 4a and now Extended Data Fig. 3k, Lines 179–180 and 243).

3. In Figure 4d (coculture experiment), there are two identical "merge", please adjust them.

We appreciate this comment. We corrected one of the "Merge" to "Merge + DIC" in Fig. 4d (Line 258).

References

- 1 Lynch, L. *et al.* Regulatory iNKT cells lack expression of the transcription factor PLZF and control the homeostasis of T(reg) cells and macrophages in adipose tissue. *Nat Immunol* **16**, 85-95, doi:10.1038/ni.3047 (2015).
- 2 LaMarche, N. M. *et al.* Distinct iNKT Cell Populations Use IFN γ or ER Stress-Induced IL-10 to Control Adipose Tissue Homeostasis. *Cell Metab* **32**, 243-258 e246, doi:10.1016/j.cmet.2020.05.017 (2020).
- 3 Cham, B. E. & Knowles, B. R. A solvent system for delipidation of plasma or serum without protein precipitation. *J Lipid Res* **17**, 176-181 (1976).
- 4 Li, V. L., Kim, J. T. & Long, J. Z. Adipose Tissue Lipokines: Recent Progress and Future Directions. *Diabetes* **69**, 2541-2548, doi:10.2337/dbi20-0012 (2020).
- 5 Munoz-Rojas, A. R. & Mathis, D. Tissue regulatory T cells: regulatory chameleons. *Nat Rev Immunol* **21**, 597-611, doi:10.1038/s41577-021-00519-w (2021).
- 6 Gause, W. C., Wynn, T. A. & Allen, J. E. Type 2 immunity and wound healing: evolutionary refinement of adaptive immunity by helminths. *Nat Rev Immunol* **13**, 607-614, doi:10.1038/nri3476 (2013).
- 7 Kabat, A. M. *et al.* Resident T(H)2 cells orchestrate adipose tissue remodeling at a site adjacent to infection. *Science immunology* **7**, eadd3263, doi:10.1126/sciimmunol.add3263 (2022).
- 8 Cao, S. *et al.* EGFR-mediated activation of adipose tissue macrophages promotes obesity and insulin resistance. *Nat Commun* **13**, 4684, doi:10.1038/s41467-022-32348-3 (2022).
- 9 Cottam, M. A., Caslin, H. L., Winn, N. C. & Hasty, A. H. Multiomics reveals persistence of obesity-associated immune cell phenotypes in adipose tissue during weight loss and weight regain in mice. *Nat Commun* **13**, 2950, doi:10.1038/s41467-022-30646-4 (2022).
- 10 Sanderson, M. P., Dempsey, P. J. & Dunbar, A. J. Control of ErbB signaling through metalloprotease mediated ectodomain shedding of EGF-like factors. *Growth Factors* **24**, 121-136, doi:10.1080/08977190600634373 (2006).
- 11 Cui, J. *et al.* Requirement for Valpha14 NKT cells in IL-12-mediated rejection of tumors. *Science* **278**, 1623-1626, doi:10.1126/science.278.5343.1623 (1997).
- 12 Vandanmagsar, B. *et al.* The NLRP3 inflammasome instigates obesity-induced inflammation and insulin resistance. *Nat Med* **17**, 179-188, doi:10.1038/nm.2279 (2011).

REVIEWERS' COMMENTS

Reviewer #1 (Remarks to the Author):

I am generally happy with the response to the questions. The authors have put in substantial efforts to improve the manuscript. A couple remaining queries/thoughts.

Regarding Figure 5, Areg seems to be found only transcriptionally. In the rebuttal in a reviewers only figure, the authors stained adipose iNKT cells from 12-week HDF mice, but there appears that there is very little expression of Areg in iNKT-17. The proportions of AREG+ cells are a bit difficult to see in the histograms. Under what conditions is Areg protein expressed in iNKT17 cells or found secreted in supernatant? Is the expression of Areg affected/increased by TCR stimulation? I also don't see Areg as a differentially expressed gene in NKT-17 of thymus vs adipose (Sup table3). I feel the wording in the abstract is too strong to say that Areg is secreted by adipose iNKT17. I don't seem to see what is the evidence is that Areg is secreted by iNKT17? I would say they a-iNKT17 have the potential to secrete Areg, and Areg is involved in stimulating adipose stem cell proliferation.

Minor comment:

In the figure 2 legend can the authors please provide the abbreviations for As, Au, Ac.

Reviewer #2 (Remarks to the Author):

The authors have satisfactorily responded to all of my concerns. I appreciate the effort to answer the questions I raised despite the technical limitations of modifying gene expression in iNKT17 cells.

RESPONSE TO REVIEWERS' COMMENTS

Reviewer #1 (Remarks to the Author):

I am generally happy with the response to the questions. The authors have put in substantial efforts to improve the manuscript. A couple remaining queries/thoughts.

Regarding Figure 5, Areg seems to be found only transcriptionally. In the rebuttal in a reviewers only figure, the authors stained adipose iNKT cells from 12-week HDF mice, but there appears that there is very little expression of Areg in iNKT-17. The proportions of AREG+ cells are a bit difficult to see in the histograms. Under what conditions is Areg protein expressed in iNKT17 cells or found secreted in supernatant? Is the expression of Areg affected/increased by TCR stimulation? I also don't see Areg as a differentially expressed gene in NKT-17 of thymus vs adipose (Sup table3). I feel the wording in the abstract is too strong to say that Areg is secreted by adipose iNKT17. I don't seem to see what is the evidence is that Areg is secreted by iNKT17? I would say they a-iNKT17 have the potential to secrete Areg, and Areg is involved in stimulating adipose stem cell proliferation.

Minor comment:

In the figure 2 legend can the authors please provide the abbreviations for As, Au, Ac.

Reviewer #2 (Remarks to the Author):

The authors have satisfactorily responded to all of my concerns. I appreciate the effort to answer the questions I raised despite the technical limitations of modifying gene expression in iNKT17 cells.

Response to the reviewers' comments
manuscript NCOMMS-23-13999

Unique adipose invariant natural killer T cell subpopulations control adipocyte turnover in mice

Reviewer #1:

I am generally happy with the response to the questions. The authors have put in substantial efforts to improve the manuscript. A couple remaining queries/thoughts.

Regarding Figure 5, Areg seems to be found only transcriptionally. In the rebuttal in a reviewers only figure, the authors stained adipose iNKT cells from 12-week HFD mice, but there appears that there is very little expression of Areg in iNKT-17. The proportions of AREG+ cells are a bit difficult to see in the histograms. Under what conditions is Areg protein expressed in iNKT17 cells or found secreted in supernatant? Is the expression of Areg affected/increased by TCR stimulation? I also don't see Areg as a differentially expressed gene in NKT-17 of thymus vs adipose (Sup table3). I feel the wording in the abstract is too strong to say that Areg is secreted by adipose iNKT17. I don't seem to see what is the evidence is that Areg is secreted by iNKT17? I would say they a-iNKT17 have the potential to secrete Areg, and Areg is involved in stimulating adipose stem cell proliferation.

We appreciate these comments and suggestions. Also, we apologize for any inconvenience caused by difficulty seeing the histogram data. The dotted line in the previous Reviewer's only Figure was the boundary that separates AREG-positive cells from negative cells, and cells that exceed this boundary were considered AREG-positive cells. Quantitative analysis of AREG-positive cells indicated that the proportion of AREG-positive cells was similar between iNKT17, Treg, and ILC2. It could be seen that there seemed to be low expression of Areg in iNKT17 cells because the total number of iNKT17 cells was relatively much smaller than the other two such as Treg and ILC2.

To determine whether AREG is increased by TCR stimulation, we isolated adipose tissue stromal vascular fractions (SVFs) from 12-week HFD-fed male mice and activated SVFs with PMA/Ionomycin which mimics TCR stimulation in the presence of Brefeldin A which traps cytokine inside the cells. Then, we analyzed them by flow cytometry with intracellular cytokine staining of AREG. As the 'Unstimulated' histogram in the below Reviewer's only Figure 1a represents adipose SVF which was treated only with Brefeldin A without PMA/Ionomycin, we concluded that AREG expression would be induced by TCR stimulation (PMA/Ionomycin).

As you commented, *Areg* is not differentially expressed in between thymic iNKT17 and adipose iNKT17 cells. Thus, we have revised our Abstract with the following sentence according to your suggestion.

"Furthermore, adipose iNKT cells have the potential to secrete AREG, and AREG is involved in stimulating adipose stem cell proliferation."

Reviewer's only Figure 1 from the previous Response to the reviewer file

a, The ratio of AREG-positive cells among indicated cell types and representative FACS plot of intracellular cytokine staining. Adipose tissue stromal vascular fraction (SVF) from 12-week HFD-fed male mice was incubated in complete RPMI media containing Brefeldin A (10 µg/ml) with or without PMA (100 ng/ml) and Ionomycin (1 µg/ml) for 5 hours. Trapped AREG proteins inside the cells were stained and analyzed by FACS in A-iNKT17 cells, Treg cells, and ILC2 (n = 4). **b**, Gating strategy for A-iNKT17 cells, Treg cells, and ILC2 in adipose tissue SVF. Data are represented as mean ± SD.

Minor comment:

In the figure 2 legend can the authors please provide the abbreviations for As, Au, Ac.

According to this, we added abbreviations for As-iNKT1 and Au-iNKT1 at the end of the Figure 2 legend. Since Ac-iNKT1 cells are not mentioned in Figure 2, we added an abbreviation of Ac-iNKT1 cells at the end of Figure 3 legend.

Reviewer #2:

The authors have satisfactorily responded to all of my concerns. I appreciate the effort to answer the questions I raised despite the technical limitations of modifying gene expression in iNKT17 cells.

We deeply appreciate this positive review.